# ON THE SCALING THEORY OF MULTI-LAYER TRANSFORMERS

## ABSTRACT

The scaling law, a cornerstone of Large Language Model (LLM) development, predicts improvements in model performance with increasing computational resources. Yet, while empirically validated, its theoretical underpinnings remain poorly understood. This work formalizes the learning dynamics of transformer-based language models as an ordinary differential equation (ODE) system, then approximates this process to kernel behaviors. Departing from prior toy-model analyses, we rigorously analyze one-pass stochastic gradient descent (SGD) training for multi-layer transformers on sequence-to-sequence data with arbitrary data distribution, closely mirroring real-world conditions. Our analysis characterizes the convergence of generalization error to the irreducible risk as computational resources scale with data. We derive an excess risk of $\Theta(C^{-1/8})$ for computational cost $C$. The theory reveals a phase transition: under specific conditions, the generalization risk's upper bound drops sharply to $\exp(-C^{1/4})$ before reverting to its original decay rate. This transition delineates three scaling regimes—*classical, over-parameterization, and data-limited*—which we analyze for their impact on scaling efficiency and the emergence of grokking.

## 1 INTRODUCTION

The emergence of transformer-based Large Language Models (LLMs) such as ChatGPT (ChatGPT, 2022), GPT-4 (Achiam et al., 2023; Bubeck et al., 2023), DeepSeek (Bi et al., 2024; Liu et al., 2024b;a; Grattafiori et al., 2024), and LLaMA (Touvron et al., 2023a;b; Grattafiori et al., 2024) has driven transformative advancements across multiple domains (Brown et al., 2020). Tasks like code generalization (Li et al., 2023; Guo et al., 2024), conversational systems (Maaz et al., 2023; Xu et al., 2023; Zheng et al., 2024), and mathematical reasoning (Hendrycks et al., 2020; Yu et al., 2023a; Yao et al., 2023), once considered exclusive to human expertise, are now routinely mastered by these AI systems.

This remarkable progress is fundamentally tied to computational scaling. Empirical evidence reveals a consistent pattern: as the compute budget for optimally training and deploying language model increases, their demonstrated intelligence scales correspondingly (Kaplan et al., 2020; Hoffmann et al., 2022). This phenomenon has been systematically categorized into the following principle:

> **Scaling Law** (Kaplan et al., 2020; Hoffmann et al., 2022): Model capabilities improve predictably with increased computational investment during training, achieved through three-dimensional scaling: model parameter count, training duration, and dataset size.

While the scaling law has been extensively validated through empirical research, a critical gap persists in its theoretical foundation. Current literature lacks a mathematically rigorous framework to explain why these principles exhibit such predictable improvements in model performance, or to systematically justify their reliability in guiding the development of massive-scale transformer architectures (Lin et al., 2024). Existing theoretical studies on scaling laws have predominantly focused on traditional statistical models and simplified neural architectures. Key investigations include analyses of shallow attention networks (Lyu et al., 2025), linear regression (Lin et al., 2024; Daliri et al., 2024), kernel regression (Chen et al., 2025), and data selection methodologies (Jain et al., 2018), among others. However, these works lack generalizability to modern deep learning systems. Notably, the scaling theory underpinning transformer-based LLMs (Vaswani et al., 2017) — the dominant paradigm in contemporary AI — remains largely unexplored.

Our work addresses this gap by establishing theoretical foundations for computational scaling effectiveness in training multi-layer transformer-based language models on sequence-to-sequence data. Specifically, we derive a quantifiable relationship between computational expenditure and model capabilities, formulating an optimized error bound for training objectives that explicitly depends on allocated computational resources. In particular, the goal of this work is to address the following critical question:

*What are the theoretical limits for computational resource allocation to ensure the convergence of the generalization error bound during transformer-based language model training?*

We present the first comprehensive analysis of the training dynamics of transformer-based language models using the one-pass Stochastic Gradient Descent (SGD) algorithm, along with a convergence guarantee for arbitrary training error. This topic has been largely unexplored due to the inherent complexity of attention mechanisms, the multi-layered structure of transformers, and the extensive matrix computations required for sequence-to-sequence data processing. To address these challenges, we adopt a kernel-based analytical framework to investigate the scaling behavior. By analyzing this scenario, we establish the lower and upper bounds for the expected risk on the whole data distribution. Overall, we make the following contributions:

- We first simplify the complicated matrix computation to the parallel vector computation utilizing the decoder-only property of a generative transformer-based language model. Therefore, we formalize its explicit learning dynamics, which we assume the learning behavior of LLMs is constrained to the kernel regimes due to the over-parameterization. This is referred to the *Lazy Learning* (Jacot et al., 2018; Du et al., 2019), where the model merely memorizes all data points during optimization. (See Section 4)

- We further explore several benefits that emerge under the trend of model scaling, e.g., converging kernel perturbation, which means that once kernel behavior exists at initialization, it will remain stable during training. Moreover, we demonstrate how the training convergence rate exponentially improves with linearly increasing model depth. Finally, we showcase the guaranteed training convergence and approximation bound related to vital scaling factors (model size, dataset size, training time, compute cost). (See Section 5)

- The main contribution of this work is a three-stage upper bound on the in-distribution expected risk, delineating the scaling process into the *classical stage*, the *over-parameterization stage*, and the *data-limited stage*. Our theoretical analysis shows that the generalization error initially decreases at a high rate with reducing scaling profit (i.e., inefficient), then enters a period of slower error reduction but growing scaling profit (i.e., highly efficient), which aligns with the empirically observed grokking phenomenon (Power et al., 2022). Finally, the convergence rate diminishes again in the *data-limited stage*. We provide theoretical guarantees for this risk landscape and present supporting experimental results. (See Section 6 and Section 7)

## 2 RELATED WORK

**Scaling Law.** Several recent works have empirically explored scaling laws in deep neural networks (Kaplan et al., 2020; Hoffmann et al., 2022; Rosenfeld et al., 2021; Hestness et al., 2017; Rosenfeld et al., 2019). The study of neural scaling laws can also be traced back to earlier foundational works (Caponnetto & De Vito, 2007; Steinwart et al., 2009; Ahmad & Tesauro, 1988). From a theoretical perspective, various solvable models have been developed using random feature models (Bahri et al., 2024; Atanasov et al., 2021; 2024; Bordelon et al., 2024; Paquette et al., 2024) to analyze neural scaling laws under specific limits. Additionally, theoretical analyses on linear models (Wei et al., 2022a; Bordelon et al., 2020; Seleznova & Kutyniok, 2022; Bordelon & Pehlevan, 2021; Lin et al., 2024; Lyu et al., 2025) have significantly advanced our understanding of scaling laws. In contrast to these studies, our work focuses on the sequence-to-sequence stochastic training of multi-layer transformer-based language models, a topic that has not been widely discussed in prior research.

**Neural Tangent Kernel and Learning Theory.** The Neural Tangent Kernel (NTK), introduced by Jacot et al. (2018), has become a foundational framework for understanding the gradient flow of neural networks during training. It reveals that, in the infinite-width limit, neural networks are equivalent to Gaussian processes at initialization. This equivalence has been extensively studied in numerous works (Li & Liang, 2018; Du et al., 2019; Song & Yang, 2019; Allen-Zhu et al., 2019; Wei et al.,

2019; Bietti & Mairal, 2019; Lee et al., 2020; Chizat & Bach, 2020; Shi et al., 2021; Zhou et al., 2021; Seleznova & Kutyniok, 2022; Gao et al., 2023; Li et al., 2024; Shi et al., 2024b), which highlight the robust performance and learning capabilities of over-parameterized neural networks. The NTK framework has gained popularity for its ability to elucidate the emerging abilities of large-scale neural networks. Notable advancements include the introduction of the Convolutional NTK (CNTK) by Arora et al. (2019), the Recurrent NTK by Alemohammad et al. (2020), and the concept of infinite attention via NNGP and NTK for attention networks by Hron et al. (2020). Furthermore, Malladi et al. (2023) examined the training dynamics of fine-tuning LLMs using NTK, demonstrating its efficiency in optimizing these systems. These contributions underscore the growing importance of NTK in the theoretical analysis of modern neural networks.

**Science of Transformer-based Language Models.** The complex architecture and stochastic optimization processes of transformer-based language models pose significant challenges for theoretical analysis. However, developing theoretical guarantees for LLMs is essential for advancing their design and performance. Recent research has addressed various aspects of LLMs, including improving efficiency (Alman & Song, 2023; 2024a;b; Han et al., 2024; Kacham et al., 2023; Addanki et al., 2023; Deng et al., 2024; Shi et al., 2024a), optimizing training processes (Deng et al., 2023; Li et al., 2024), analyzing "white-box" transformers (Yu et al., 2023b;c; Ferrando et al., 2024; Pai et al., 2024), and investigating the emergent abilities of LLMs (Brown et al., 2020; Wei et al., 2022b; Allen-Zhu & Li, 2023a;b;c; 2024). By bridging theoretical understanding with practical advancements, these studies provide valuable insights for the development of the next generation of AI systems.

## 3 PRELIMINARY

This section provides the preliminary for our analysis, where we introduce some basic notations in Section 3.1, and present the problem setup in Section 3.2. We encourage the reader to refer to Appendix B for the formal technical preliminary.

### 3.1 BASIC NOTATIONS

Let $[d] = \{1, 2, \ldots, d\}$. For $u \in \mathbb{R}^d$, the $\ell_p$-norm is $\|u\|_p := (\sum_{k=1}^d |u_k|^p)^{1/p}$. The Frobenius norm of $U \in \mathbb{R}^{d_1 \times d_2}$ is $\|U\|_F := (\sum_{(k_1, k_2) \in [d_1] \times [d_2]} U_{k_1, k_2}^2)^{1/2}$. For a matrix $A \in \mathbb{R}^{d \times d}$, $\lambda_{\min}(A)$ denotes its smallest eigenvalue. The indicator function $\mathbb{I}\{E_1, \ldots, E_n\}$ equals 1 if all events $E_1, \ldots, E_n$ occur, and 0 otherwise. The mapping $\mathrm{mat} : \mathbb{R}^{d^2} \to \mathbb{R}^{d \times d}$ reshapes a vector $a$ into a matrix such that $\mathrm{mat}_{k_1, k_2}(a) = a_{(k_1-1)d+k_2}$ for $(k_1, k_2) \in [d] \times [d]$. Conversely, $\mathrm{vec} : \mathbb{R}^{n \times d} \to \mathbb{R}^{nd}$ flattens a matrix $A$ into a vector with $\mathrm{vec}_k(A) = A_{\lfloor k/d \rfloor, \, k - \lfloor k/d \rfloor \cdot d}$ for $k \in [nd]$. For a function $f : X \to \mathbb{R}^{d_1 \times d_2}$, $f_{k_1, k_2}(x)$ denotes the $(k_1, k_2)$-entry of $f(x)$. $e_k \in \mathbb{R}^d$ is the standard basis vector with a 1 in the $k$-th entry and 0 elsewhere. Finally, we denote $a \wedge b := \max\{a, b\}$ for $a, b \in \mathbb{R}$.

### 3.2 SETUPS

**Data Distribution.** We consider a sequence-to-sequence regression task with an input space $\mathcal{X} \subseteq \mathbb{R}^{L \times d}$ as the space of encoded input sequence[1], and $\mathcal{Y} \subseteq [C_{\mathrm{lower}}, C_{\mathrm{upper}}]^{L \times d}$ for constant $C_{\mathrm{lower}}, C_{\mathrm{upper}} \in \mathbb{R}$ is the space of encoded target output. We denote $F^*$ as the optimal measurable function mapping $\mathcal{X} \to \mathcal{Y}$ with minimum risk. Given a model class $\mathcal{F}$, then for the distribution $\mathcal{D} = \{(X, F^*(X) + \Xi), \Xi \in \mathbb{R}^{L \times d} \text{ is some random noise}\} \subset \mathcal{X} \times \mathcal{Y}$ and model function $F \in \mathcal{F}$, the expected risk (we consider as the generalization error bound) and excess risk of $F$ is defined as:

$$\text{Expected Risk: } \mathcal{R}(F) := \mathbb{E}_{(X,Y) \sim \mathcal{D}}[\|F(X) - Y\|_F^2], \quad \text{Excess Risk: } \Delta\mathcal{R}(F) := \mathcal{R}(F) - \mathcal{R}(F^*).$$

Besides, we have an accessible dataset $\mathbb{D} = \{(X_i, Y_i)\}_{i=1}^n \subset \mathcal{D}$ where each data point independently and uniformly sampled from $\mathcal{D}$. The random noise $\Xi \in \mathbb{R}^{L \times d}$ is centered by $\mathbf{0}_{L \times d}$. For any input matrix $X \sim \mathcal{X}$, $\|X_\ell\|_2 = \Theta(1)$ holds for each token vector $\ell \in [L]$ due to the utilization of RMS normalization (Zhang & Sennrich, 2019) after the embedding layer.

**Model Function.** The standard transformer architecture introduced in Vaswani et al. (2017) integrates multiple self-attention layers with token-wise feed-forward layers. The fundamental

---

[1] We choose the max length of sequence $L$ considerably large, for sequences with a length less than $L$, we use padding to fill them.

architecture, decoder-only transformers (Radford et al., 2019), processing a sequence of $L$ tokens, each represented by a $d$-dimensional embedding vector, which are compactly arranged into a matrix $X \in \mathbb{R}^{L \times d}$. An $N$-layer transformer model is formally defined as:

$$F(X, \theta) := \varepsilon \cdot F_{(N)}(F_{(N-1)}(\cdots F_{(2)}(F_{(1)}(X + E, \theta), \theta) \cdots), \theta), \tag{1}$$

where $E \in \mathbb{R}^{L \times d}$ is the positional embedding matrix[2] and $\theta$ is the set of all trainable parameters. $\varepsilon > 0$ is the grokking coefficient, which we show the relationship between its value and the grokking phenomenon Power et al. (2022); Nanda et al. (2023); Liu et al. (2022) in Section 6.2. Each $F_{(\nu)}$ (for $\nu \in [N]$) represents the $\nu$-th transformer block and is given by:

$$F_{(\nu)}(X, \theta) := \frac{\omega}{\sqrt{m}} \mathrm{ReLU}\left(\mathrm{Softmax}\left(\kappa \cdot X U_{(\nu)} X^\top + M\right) X W_{(\nu)}\right) A_{(\nu)} + X,$$

where $U_{(\nu)} \in \mathbb{R}^{d \times d}$, $W_{(\nu)}, A_{(\nu)}^\top \in \mathbb{R}^{d \times m}$ are model parameters. $\kappa$ is the scaling factor of attention, $\omega$ is the scaling coefficient of output. $M \in \mathbb{R}^{L \times L}$ is the causal attention mask. We especially use $w_{(\nu),r}, a_{(\nu),r} \in \mathbb{R}^d$ to denote the $r$-th column of $W_{(\nu)}$ and the $r$-th row of $A_{(\nu)}$, respectively.

**Initialization and Training.** For every layer $\nu \in [N]$, the corresponding parameters of $F_{(\nu)}$ is denoted as $U_{(\nu)}, W_{(\nu)}, A_{(\nu)}$. Each entry of $U_{(\nu)}, W_{(\nu)}$ is initialized from the standard Gaussian distribution $\mathcal{N}(0, 1)$, we then denote them as $U_{(\nu)}(0), W_{(\nu)}(0)$. Besides, each entry of $A_{(\nu)}$ is initialized from a uniform distribution $\mathrm{Uniform}\{-1, +1\}$ and is frozen during training. The flattened vector of the whole trainable parameters is denoted as $\theta(0) \in \mathbb{R}^\mathsf{M}$ where $\mathsf{M} = N(md + d^2)$ is the number of trainable parameters.

Given the training dataset $\mathbb{D} = \{(X_i, Y_i)\}_{i=1}^n \subset \mathcal{D}$, we define the overall training objective as:

$$\mathcal{L}(t, \mathbb{D}) := \mathop{\mathbb{E}}_{(X,Y) \sim \mathbb{D}}[\|F(X, \theta(t)) - Y\|_F^2].$$

Therefore, we consider a combination of the one-pass stochastic gradient descent (SGD) algorithm and *gradient flow* to update. At $t$-step optimization, we sample a unbiased subset $\mathbb{B}(t) \subseteq \mathbb{D}$ satisfying $\mathbb{E}[\mathcal{L}(t, \mathbb{B}(t))] = \mathcal{L}(t, \mathbb{D})$ and $\int_0^T \mathbb{B}(t) \mathrm{d}t = \mathbb{D}$ for training time $T > 0$. Then the ordinary differential equation (ODE) of $U_{(\nu)}(t), W_{(\nu)}(t)$ and their update rule are given by:

$$\frac{\mathrm{d}}{\mathrm{d}t} U_{(\nu)}(t) = -\frac{\mathrm{d}}{\mathrm{d}U_{(\nu)}(t)} \mathcal{L}(t, \mathbb{B}(t)), \quad \frac{\mathrm{d}}{\mathrm{d}t} W_{(\nu)}(t) = -\frac{\mathrm{d}}{\mathrm{d}W_{(\nu)}(t)} \mathcal{L}(t, \mathbb{B}(t)),$$

$$U_{(\nu)}(t + \tau) = U_{(\nu)}(t) + \int_t^{t+\tau} \frac{\mathrm{d}}{\mathrm{d}s} U_{(\nu)}(s) \mathrm{d}s, \quad W_{(\nu)}(t + \tau) = W_{(\nu)}(t) + \int_t^{t+\tau} \frac{\mathrm{d}}{\mathrm{d}s} W_{(\nu)}(s) \mathrm{d}s. \tag{2}$$

Hence, we denote the training algorithm that depends on training time and dataset size as $\mathcal{A}_{T,n}(\theta(0), \mathbb{D}) := \{\theta(0) + \int_0^T -\frac{\mathrm{d}}{\mathrm{d}\theta} \mathcal{L}(s, \mathbb{B}(s)) \mathrm{d}s, \mathbb{B}(s) \subseteq \mathbb{D}, s \in [0, T]\}$.

## 4 LEARNING DYNAMICS OF SCALING TRANSFORMERS

In this section, we provide the explicit model learning dynamics formulations layer by layer. We first introduce several key simplifications in Section 4.1 to facilitate the preliminary analysis. Thus, Section 4.2 formulates an explicit ODE of the learning dynamics of scaling multi-layer transformer-based language models.

### 4.1 SIMPLIFICATIONS

Utilizing the attention network's decoder-only property, it is obvious that the matrix computation can be parallelized to vector computation. We use $X_{i,\leq \ell} \in \mathbb{R}^{\ell \times d}$ to denote the first-$\ell$ tokens ($\forall \ell \in [L]$) of matrix $X_i$ in $\mathbb{D}$. Hence, we compact the outputs of the model function and targets on the whole dataset to two matrices $\mathsf{F}(t), \mathsf{Y} \in \mathbb{R}^{nL \times d}$, where $\mathsf{F}_{(i-1)L+\ell}(t) = F_\ell(X_i, \theta(t)) \in \mathbb{R}^d$ and $\mathsf{Y}_{(i-1)L+\ell} = Y_{i,\ell}$ for each $(X_i, Y_i)$ in training dataset $\mathbb{D}$ and $\ell \in [L]$. Therefore, we derive $\mathcal{L}(t, \mathbb{D}) = \frac{1}{n} \|\mathsf{F}(t) - \mathsf{Y}\|_F^2$ (See Lemma C.3). Besides, we list following helpful functions ($(i, \ell) \in [n] \times [L]$):

---

[2]We choose $E = \mathbf{0}_{L \times d}$ (NoPE, No Positional Embedding, (Kazemnejad et al., 2023)) or ignore it as a fixed matrix (this can be regarded as a part of the training dataset) in the range of this paper.

- (Hidden State) $\Lambda_{(\nu),i}(t) := F_{(\nu)}(\Lambda_{(\nu-1),i}(t), \theta(t)) \in \mathbb{R}^{L \times d}$ for $\nu \in [N]$, $\Lambda_{(0),i}(t) = X_i + E$.

- (Attention Scores) $\sigma_{(\nu),(i-1)L+\ell}(X) = \text{Softmax}_\ell(\Lambda_{(\nu),i}(t) U_{(\nu)}(t) \Lambda_{(\nu),i}(t)^\top + M) \in \mathbb{R}^L$.

- (Attention Output) $o_{(\nu),(i-1)L+\ell}(t) := \Lambda_{(\nu-1),i}(t)^\top \cdot \sigma_{(\nu),(i-1)L+\ell}(t) \in \mathbb{R}^d$.

- ($\ell$-th Token of Hidden State)

$$\mu_{(\nu),(i-1)L+\ell}(t) := \frac{\omega}{\sqrt{m}} \sum_{r=1}^m a_{(\nu),r} \cdot \phi(\langle o_{(\nu),(i-1)L+\ell}(t), w_{(\nu),r}(t) \rangle) \in \mathbb{R}^d,$$

  where $\phi(x) := \max\{0, x\}, \forall x \in \mathbb{R}$. $\mu_{(0),(i-1)L+\ell}(t) = X_{i,\ell} + E_\ell$.

- (Model Output) $\mathsf{F}_{(i-1)L+\ell}(t) = \varepsilon \cdot \sum_{\nu=0}^N \mu_{(\nu),(i-1)L+\ell}(t) \in \mathbb{R}^d$.

## 4.2 Key Derivation for Learning Dynamics

The primary challenge in understanding the learning dynamics of finite-deep transformers is the complex analysis of gradient flow, which differs from the study of shallow or infinite-deep neural networks. We overcome the complexity by cleverly utilizing the multivariable chain rules.

First, we define the kernel matrix at $\nu$-th layer as $H_{(\nu)} \in \mathbb{R}^{nL \times nL}$ and its $(i,j)$-th entry ($\forall (p,q) \in [nL] \times [nL]$) is defined as:

$$H_{(\nu),p,q}(t) := \underbrace{\langle \beta_{(\nu),p}(t), \beta_{(\nu),q}(t) \rangle}_{\text{kernel w.r.t. } W_{(\nu)}(t)} + \underbrace{\langle \gamma_{(\nu),p}(t), \gamma_{(\nu),q}(t) \rangle}_{\text{kernel w.r.t. } U_{(\nu)}(t)},$$

Here, we let:

$$\beta_{(\nu),p}(t) := \frac{\omega}{\sqrt{m}} \underbrace{o_{(\nu),p}(t)}_{d \times 1} \otimes \underbrace{\mathbf{1}_{W_{(\nu)}(t)^\top o_{(\nu),p}(t) > 0}}_{m \times 1} \in \mathbb{R}^{md},$$

$$\gamma_{(\nu),p}(t) := \frac{\omega \cdot \kappa}{\sqrt{m}} \underbrace{(\Lambda_{(\nu-1),i,\ell,*}(t) \otimes \Lambda_{(\nu-1),i}(t))^\top}_{d^2 \times L} \underbrace{(\text{diag}(\sigma_{(\nu),p}(t)) - \sigma_{(\nu),p}(t)\sigma_{(\nu),p}(t)^\top)}_{L \times L}$$

$$\underbrace{\Lambda_{(\nu-1),i}(t)}_{L \times d} \sum_{r \in [m]} \underbrace{w_{(\nu),r}(t) \, \mathbb{I}\{o_{(\nu),p}(t)^\top w_{(\nu),r}(t) > 0\}}_{d \times 1} \in \mathbb{R}^{d^2},$$

where $\otimes$ is the Kronecker product and $i = \lfloor p/L \rfloor$, $\ell = p \bmod L$. The indicator vector $\mathbf{1}_{W_{(\nu)}(t)^\top o_{(\nu),p}(t) > 0} \in \{0,1\}^m$ where its $r$-th entry is $\mathbb{I}\{(W_{(\nu)}(t)^\top o_{(\nu),p}(t))_r > 0\}$ for $r \in [m]$. The layer-wise training dynamics are thereby shown as the following lemma:

**Lemma 4.1** (Learning dynamics, informal version of Lemma C.6). *The learning dynamics of the multi-layer transformer Eq. (1) is given by:*

$$\mathbb{E}[\frac{\mathrm{d}}{\mathrm{d}t} \mathcal{L}(t, \mathbb{D})] = - \sum_{\nu \in [N]} \underbrace{\text{vec}\left(\frac{\mathrm{d}}{\mathrm{d}\mu_{(\nu)}(t)} \mathcal{L}(t, \mathbb{D})\right)^\top}_{1 \times nLd} \cdot \underbrace{(H_{(\nu)}(t) \otimes I_d)}_{nLd \times nLd} \cdot \underbrace{\text{vec}\left(\frac{\mathrm{d}}{\mathrm{d}\mu_{(\nu)}(t)} \mathcal{L}(t, \mathbb{D})\right)}_{nLd \times 1}$$

*where $\mu_{(\nu)}(t)$ is a $nL \times d$ matrix, $\mu_{(\nu),p}(t)$ is the $(p \bmod L)$-th row of $v$-th layer output regarding to input matrix $X_{\lfloor p/L \rfloor}$ for any $p \in [nL]$ and $\nu \in [N]$.*

*Proof sketch of Lemma 4.1.* Although the derivation of the gradient is complicated, the technique used for the proof is just the chain rule. We provide the complete proof in Appendix C. ☐

Lemma 4.1 sums contributions from all layers $\nu \in [N]$, providing a powerful decomposition. This granular view suggests that each layer independently contributes to minimizing the loss based on its own kernel and local gradient. Different layers may learn at varying speeds or contribute differently to the overall task, depending on their respective kernel structures.

## 5 TRAINING CONVERGENCE AND APPROXIMATION GUARANTEES

In this section, we showcase the training convergence with an arbitrary error by limiting the kernel matrix $H'_{(\nu)}(0)$ at initialization to the Neural Tangent Kernel (NTK) (Jacot et al., 2018; Du et al., 2019) assumption. We state the formal assumption and the basic inductive proof in Section 5.1, and Section 5.2 demonstrates the results of the training convergence.

### 5.1 ASSUMPTIONS AND INDUCTIONS

Following the setting of Du et al. (2019) (see Assumption 3.1), we align this mild assumption in our analysis as shown:

**Assumption 5.1.** *Defining $H'_{(\nu)}(0) \in \mathbb{R}^{nL \times nL}$ where its $(i,j)$-th entry ($\forall (i,j) \in [nL] \times [nL]$) is given by $H'_{(\nu),i,j}(0) := \langle \beta_{(\nu),i}(t), \beta_{(\nu),j}(t) \rangle$. For all $\nu \in [N]$, we assume $\frac{1}{\omega} H'_{(\nu)}(0)$ is positive definite (PD), formally, $\lambda_{(\nu)} := \lambda_{\min}(\frac{1}{\omega} H'_{(\nu)}(0)) > 0$.*

**Converging Kernel Perturbation and Kernel-Based Lazy Learning.** Assuming $\lambda_{\min}(H'_{(\nu)}(0)) > 0$, we extend this PD property to the case during the training and the case of $H_{(\nu)}(t)$ that requires bounded summational update of weight at an arbitrary positive time $t > 0$. This connects to the *Lazy Learning* regime (Jacot et al., 2018; Du et al., 2019), where the updates of weights are limited in some high-dimensional ball with radius $R$. We first give the *Good Properties* requirements for provable arbitrary convergence below:

**Definition 5.2** (Good Properties and Good Model Class). *We fix the dataset size $n$, then we say model $F(X, \theta)$ has Good Properties once it satisfies (constant $C > 0$):*

*1. $\omega = o(\frac{1}{NL^2 d^{2.5} B^3})$;*    *2. $\kappa = \frac{1}{\sqrt{m}}$;*    *3. $m = \Omega(\frac{n^3 L^5 \exp(Cd)}{\varepsilon^6 \omega^6 \lambda^6 \delta^3 N^2})$.*

*Good Model Class is $\mathcal{F}_{\mathsf{M,T,N}}(\mathbb{D}) := \{F(\cdot, \theta(\mathsf{T})), \theta(0) \sim \mathcal{N}(0, I_\mathsf{M}), \theta(\mathsf{T}) \in \mathcal{A}_{\mathsf{T,N}}(\theta(0), \mathbb{D})\}$ for any $\mathbb{D} \subset \mathcal{D}$ and $|\mathbb{D}| = \mathsf{N}$.*

**Lemma 5.3** (Informal version of Lemma E.1). *Assuming Assumption 5.1 and Definition 5.2 hold, denote the failure probability $\delta \in (0, 0.1)$, then the kernel perturbation bound is: $\Pr\left[\lambda_{\min}(H_{(\nu)}(t)) < \lambda/2\right] < \delta$. Therefore, bounding loss dynamics is given by ($C > 0$ is some constant): $\Pr\left[\mathbb{E}[\frac{\mathrm{d}}{\mathrm{d}t}\mathcal{L}(t, \mathbb{D})] > -C \cdot \omega\lambda N \cdot \mathcal{L}(t, \mathbb{D})\right] < \delta$.*

*Proof sketch of Lemma 5.3.* The stability of the PD property is ensured by $\lambda_{\min}(H') \geq \lambda_{\min}(H) - \|H - H'\|_F$ (Fact B.11) since the considerably small perturbation $\|H - H'\|_F$. The upper bound on perturbation converges due to increasing feed-forward layer width $m$ and decreasing weight perturbation radius $R$, leading to the *Lazy Learning* regime. The complete proof of this lemma is stated in Appendix E. □

### 5.2 RESULTS: CONVERGENCE OF KERNEL REGIMES

**Factors of Scaling Law.** To begin with, we list several crucial factors of the scaling law below. The number of training time $\mathsf{T}$ is ensured by the dataset size and the minimum batch size $|\mathbb{B}|$ due to the one-pass SGD. In addition, the model size refers to the number of trainable parameters in our model, which could be trivially calculated, and the definition of the total compute holds as our compute analysis in Lemma F.1.

**Definition 5.4.** *We define:*

- *Model Size:* $\mathsf{M} := O(N(md + d^2)) = O(Nmd)$.
- *Training Time:* $\mathsf{T} := \frac{\mathsf{N}}{|\mathbb{B}|}$.
- *Dataset Size:* $\mathsf{N} := n$.
- *Total Compute:* $\mathsf{C} := O(\mathsf{MN})$.

**Training Convergence and Approximation.** The convergence guarantee of implementing Eq (2) as the continuous-time optimization is stated below:

**Theorem 5.5** (Informal version of Theorem F.2). *Let all scaling law factors be defined as Definition 5.4 and Assumption 5.1 and Definition 5.2 hold. Denote the failure probability $\delta \in (0, 0.1)$, $\alpha_{\text{approx}} = \text{poly}(\exp(d), L, n, \frac{1}{\delta})$. For the Good Model Class $\mathcal{F}_{\mathsf{M},\mathsf{T},\mathsf{N}}(\mathbb{D})$, with a probability at least $1 - \delta$, we have: $\mathcal{L}(t, \mathbb{D}) = \mathbb{E}_{(X,Y) \sim \mathcal{D}}[\|F(X) - Y\|_F^2] \leq \exp(-\frac{\varepsilon}{\alpha_{\text{approx}}}\mathsf{MT}), \forall F \in \mathcal{F}_{\mathsf{M},\mathsf{T},\mathsf{N}}(\mathbb{D}).$*

*Proof Sketch of Theorem 5.5.* Firstly, we confirm the connection between hidden-state gradient norm and the training objective in Part 15 of Lemma D.1, the model convergence therefore exponentially benefits from the model size (neural depth and width) and the training time. Besides, the variance produced by the stochastic algorithm is provably reduced with a considerably large $m$. See Appendix F.2 for the detailed proofs. □

**Corollary 5.6** (Informal version of Corollary F.3). *Assuming we have arbitrary dataset size $\mathsf{N} \in (0, +\infty)$. Let all scaling law factors be defined as Definition 5.4 and Assumption 5.1 and Definition 5.2 hold. Denote the failure probability $\delta \in (0, 0.1)$. We define $\alpha = O(Ld\sqrt{\log(1/\delta)})$. For the Good Model Class $\mathcal{F}_{\mathsf{M},\mathsf{T},\mathsf{N}}(\mathbb{D}')$ with some $\mathbb{D}' \subset \mathcal{X} \times \mathcal{Y}$ and any function $F' : \mathcal{X} \to \mathcal{Y}$, arbitrary error $\epsilon > 0$ and compute cost $\mathsf{C}$, with a probability at least $1 - \delta$, we have:*

$$\inf_{F \in \mathcal{F}_{\mathsf{M},\mathsf{T},\mathsf{N}}(\mathbb{D}'), \mathbb{D}' \subset \mathcal{X} \times \mathcal{Y}} \mathbb{E}_{(X,Y) \sim \mathcal{D}}[\|F(X) - F'(X)\|_F^2] \leq \epsilon,$$

*where $\mathsf{C} = O(\mathsf{MN}) = \Omega(256\alpha^8\epsilon^{-8} \wedge \varepsilon^{-8}\omega^4 \log(2\epsilon^{-1})^4)$, $\mathsf{M} = \Omega(\mathsf{N}^3)$, $\mathsf{T} = \mathsf{N}$.*

*Proof Sketch of Corollary 5.6.* We let $\mathbb{D}' := \{(X_i, F'(X_i)), X_i \sim \mathcal{X}\}_{i=1}^{\mathsf{N}}$, then $\mathcal{F}_{\mathsf{M},\mathsf{T},\mathsf{N}}(\mathbb{D}') = \{F(\cdot, \theta(\mathsf{T})), \theta(0) \sim \mathcal{N}(0, I_\mathsf{M}), \theta(\mathsf{T}) \in \mathcal{A}_{\mathsf{T},\mathsf{N}}(\theta(0), \mathbb{D}')\}$, where we define $\mathcal{A}_{\mathsf{T},\mathsf{N}}(\theta(0), \mathbb{D}') := \{\theta(0) + \int_0^T -\frac{d}{d\theta}\mathcal{L}(s, \mathbb{B}(s))ds, \mathbb{B}(s) \subseteq \mathbb{D}'\}$. Thus, we are able to obtain the same bound in Theorem 5.5 of the model to approximate the optimal mapping $F^*$. We combine the Hoeffding inequality with the convergence rate to obtain the results trivially. The formal proof is in Appendix F.3. □

# 6 SCALING LAW

This section formally analyzes the generalization error bound of transformer-based language models. In particular, Section 6.1 showcases the general scaling law, which describes how the generalization error bound converges with the growing total computational cost. Moreover, since Theorem 6.1 demonstrates the three-stage upper bound on the in-distribution expected risk, in Section 6.2, we analyze how each variable affects the generalization in different scaling phases.

## 6.1 GENERAL PRETRAINING SCALING LAW

**Excess Risk with Optimal Dataset Size.** We state the upper bound on excess risk $\Delta\mathcal{R}(F)$ below:

**Theorem 6.1** (Informal version of Theorem G.4). *Let all pre-conditions hold as Corollary 5.6. Let arbitrary grokking coefficient $\varepsilon \in (0, 1)$, we provide the main criteria for determining scaling phase,*

$$\varepsilon \geq \mathcal{T}(\mathsf{C}) := \sqrt{\frac{\log(\mathsf{C}^{\frac{1}{8}}/\alpha)}{\mathsf{C}^{\frac{1}{4}}\omega}}. \tag{3}$$

*Hence, with a probability at least $1 - \delta$, there exists:*

$$\inf_{\mathcal{F}_{\mathsf{M},\mathsf{T},\mathsf{N}}(\mathbb{D})} \sup_{\mathbb{D} \in \mathcal{D}} \Delta\mathcal{R}(F) \leq \mathsf{R}(\mathsf{C}) := \begin{cases} O(\frac{\alpha^2}{\omega\mathsf{C}^{\frac{1}{8}}}) + O(\frac{\varepsilon \cdot d \cdot \log(1/\sqrt{\varepsilon})}{\mathsf{C}^{\frac{1}{4}}}) + \varepsilon^{\frac{3}{2}}, & \textit{Eq. (3) holds} \\ \exp(-\varepsilon^2\omega\mathsf{C}^{\frac{1}{4}}) + O(\frac{\varepsilon \cdot d \cdot \log(1/\sqrt{\varepsilon})}{\mathsf{C}^{\frac{1}{4}}}) + \varepsilon^{\frac{3}{2}}, & \textit{otherwise} \end{cases},$$

*note that $\mathsf{M} = \Omega(\mathsf{N}^3)$,       $\mathsf{T} = \mathsf{N}$,      $\mathsf{C} = O(\mathsf{MN}) = \Omega(\mathsf{N}^4)$.*

*Proof Sketch of Theorem 6.1.* This proof utilizes the results in Lemma 11 of Schmidt-Hieber (2017) for the generalization error and Corollary 5.6 for the approximation error. The full proof is shown in Appendix G.2. □

**Lower Bound.** Besides, we also give the lower bound on the excess risk as follows:

**Theorem 6.2** (Informal version of Theorem G.5). *Let all pre-conditions hold as Theorem 6.1. For any choice of $\varepsilon$, we let $\mathsf{C} = \Omega(\varepsilon^{-12})$ to offset the negative effect of the grokking coefficient $\varepsilon$, thus,*

$$\Pr\left[\inf_{\mathcal{F}_{\mathsf{M},\mathsf{T},\mathsf{N}}(\mathbb{D})} \sup_{\mathbb{D} \in \mathcal{D}} \Delta\mathcal{R}(F) \asymp O(\frac{\alpha}{\mathsf{C}^{\frac{1}{8}}})\right] \geq 1 - \delta.$$

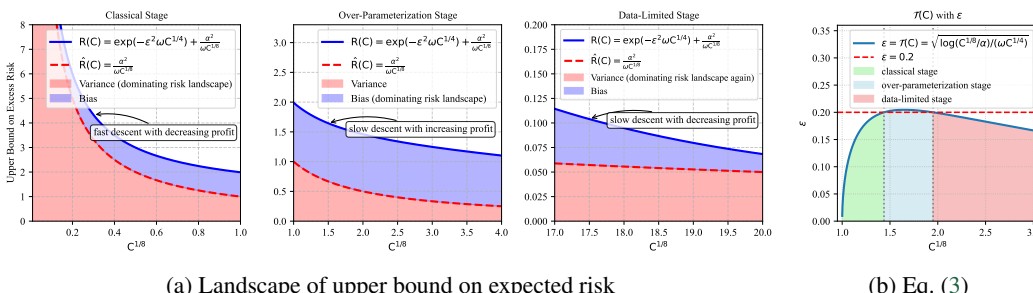

(a) Landscape of upper bound on expected risk          (b) Eq. (3)

Figure 1: (a) Visualization of the upper bound in Theorem 6.1, comparing generalization error, convergence speed, and scaling profit (defined in Section 6.2) across three scaling phases: *Classical stage* (left): Generalization error decreases rapidly, but scaling profit diminishes. *Over-parameterization stage* (center): Scaling profit increases, reflecting improved generalization efficiency per unit compute; the bias term dominates the error bound. *Data-limited stage* (right): The variance term dominates, leading to slow error reduction and declining scaling profit. (b): Visualization of Eq. (3), illustrating how compute cost $C$ and the grokking coefficient $\varepsilon$ influence the three-stage generalization behavior.

*Proof Sketch of Theorem 6.2.* Since we have no additional assumption on the data distribution, we could only obtain the lower bound using some trivial concentration inequalities (please refer to Appendix G.3 for the formal derivation). □

## 6.2 ANALYSIS ON THREE-STAGE BOUND

**Discussion of Fix $\varepsilon$ situation (Three-Stage Bound).** We define three key quantities for our analysis: (i) the generalization error bound $R(C)$ from Theorem 6.1, (ii) the generalization speed $\frac{d}{dC}R(C)$, and (iii) the scaling profit $\frac{d}{dC}(1/R(C))$, which captures generalization efficiency during scaling. For fixed $\varepsilon > 0$, the function $\mathcal{T}(C)$ increases on $[\alpha^8, C^*]$, where $C^*$ satisfies $\frac{d\mathcal{T}(C)}{dC} = 0$, then decreases monotonically to zero. The equation $\varepsilon = \mathcal{T}(C)$ thus has two solutions (Figure 1.b), partitioning the scaling process into three distinct stages: classical, over-parameterization, and data-limited regimes.

In the initial *classical stage*, the term $O(\alpha^2/(\omega C^{1/8}))$ dominates the upper bound. Insufficient neurons prevent perfect data fitting, causing the bound to initially converge rapidly before slowing due to increasing model complexity. Subsequently, in the *over-parameterization stage*, the major term in the bound of Theorem 6.1 is $\exp(-\varepsilon^2 \omega C^{\frac{1}{4}})$. Here, the model becomes over-parameterized to achieve a stable, low-variance solution, where the risk decreases exponentially with compute cost (model size and training duration) and bias dominates. Finally, in the *data-limited stage*, variance again governs the scaling rate owing to limited data samples—the model can easily memorize training data but requires more samples to generalize across the distribution. For more formal analysis, we encourage readers to refer to Appendix A.

**Transition from** *classical stage* **to** *over-parameterization stage*: **Grokking.** As shown in the middle of Figure 1.a, the training immediately accelerates to generalize, which perfectly matches the grokking phenomenon (Power et al., 2022; Nanda et al., 2023; Liu et al., 2022). Results in prior works (Kumar, 2024; Chizat et al., 2019; Lyu et al., 2023; Rubin et al., 2023) report that large initialization provably induces the grokking phenomenon, while in this work, it's equivalent to a large output-scaling coefficient $\varepsilon$. Therefore, we follow (b) in Figure 1 to provide two critical insights below, and we further evaluate its effectiveness in Section 7.

- Insight 1: Lower $\varepsilon$ (or smaller initialization) induces later beginning of grokking.
- Insight 2: Lower $\varepsilon$ (or smaller initialization) induces slower generalization speed (or smaller convergence rate of generalization) at the grokking stage.

## 7 NUMERICAL EVALUATIONS

In this section, we conduct experiments to evaluate our scaling theory and the analysis in Section 6.

**Setups.** We fine-tune the pretrained ViT (Vision Transformer, Dosovitskiy et al. (2020)) on classical image classification task, including Cifar-10, 100 (Krizhevsky et al., 2009), and MNIST (LeCun, 1998) datasets, to compare their loss variety with AdamW optimizer (Kingma & Ba, 2014).

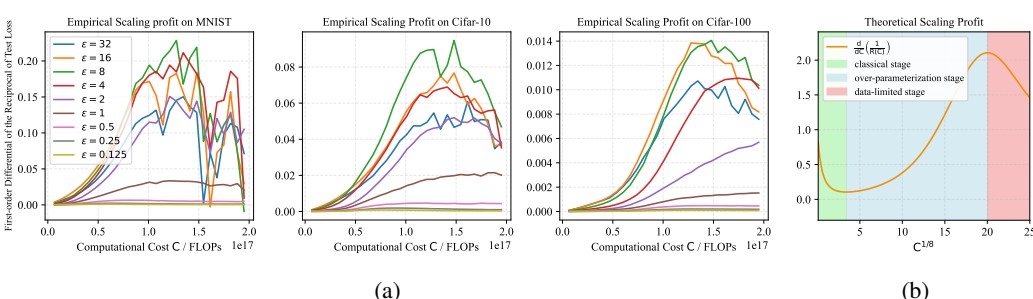

(a)          (b)

Figure 2: (a): First-order differential of the reciprocal of test error with different $\varepsilon$ and $C$ on MNIST, Cifar-10, 100 datasets. (b): Visualization of the theoretical scaling profit (see Appendix A).

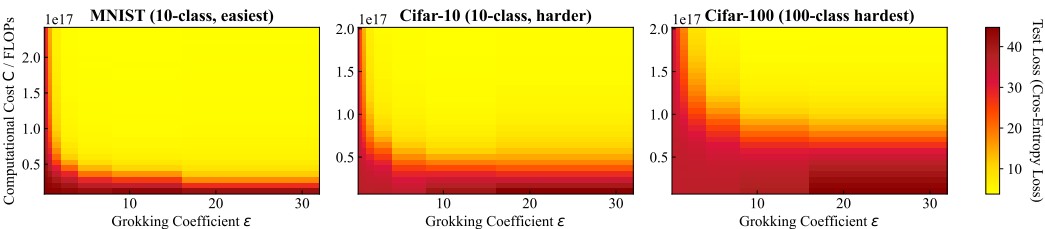

Figure 3: Results of the experiment in Section 7: Heatmaps of test loss on MNIST, Cifar-10, 100 datasets with different computational cost $C$ and grokking coefficient $\varepsilon$.

For each dataset, we sample only $1/6$ of the entire training set, taking 30 epochs of training to record the training and test cross-entropy loss curves. The value of $\varepsilon$ is iteratively chosen from $\{32, 16, 8, 4, 2, 1, 0.5, 0.25, 0.125\}$. The computational cost $C$ (computed by the total FLOPs) only depends on the training duration $T$, which is the number of total training steps in our experiments. We repeat the whole experiment 10 times for stable and fair results.

**Results of Three-Phase Scaling Law Validation.** We compute the empirical scaling profit of ViT on each dataset as the first-order derivative of the reciprocal test error with computational cost $C$ (Figure 2.1.a). The empirical curves align closely with the theoretical scaling profit within the range of $[5, 25]$ (Figure 2.b), matching and approximating the ideal shape, which further provides strong validation for our three-phase scaling law theory.

**Results of Grokking Validation.** As shown in Figure 3, as $\varepsilon$ increases, the transition region (marked by red-yellow boundaries indicative of grokking) occurs at smaller computational resource $C$, confirming Insight 1 in Section 6.2. Specifically, for smaller $\varepsilon$ (left side of each panel), these transitions emerge later or are absent during scaling. Moreover, the computational cost range associated with this transition narrows as $\varepsilon$ grows, implying that the grokking speed improves more sharply with larger $\varepsilon$ and more gradually with smaller $\varepsilon$, validating Insight 2. We also observe that larger $\varepsilon$ corresponds to higher initial risk (evidenced by darker red regions in the bottom-right) and, as task difficulty increases across the heatmaps, the onset of grokking occurs progressively later. We leave the explanation for these interesting artifacts for future direction.

## 8 CONCLUSION

This work presents a comprehensive theoretical framework for rigorously analyzing the scaling law phenomenon in LLMs, specifically addressing the empirically observed power-law relationship between model performance and computational resources from a learning theory perspective. By formalizing the dynamics of training sequence-to-sequence multi-layer transformer architectures, we establish a foundational guarantee: under allocation of compute, the generalization error of these models converges asymptotically as computational budgets increase with rate $\Theta(C^{-1/8})$ (Theorem 6.2), where $C$ is the computational cost. Furthermore, our analysis proposes a three-phase pretraining scaling theory, where the scaling process is divided by a sudden pattern transition and transition-back with improved upper bound $\exp(-C^{1/4})$. We match this regime with the grokking phenomenon, giving two insights and conduct experiments to validate its correctness.

## ETHICS STATEMENT

This is a purely theoretical paper that studies the scaling theory of training a constructed multi-layer transformer-based language model on sequence-to-sequence data distribution. To the best of our knowledge, this work provides the first analysis of the scaling law of training real-world LLMs from a learning theory perspective. As this work is theoretical and focuses on the capability of deep learning models, we don't foresee direct negative societal impacts and ethics concerns. We follow the ICLR Code of Ethics and affirm that all aspects of this research comply with the principles of fairness, transparency, and integrity.

In particular, we provide our clarification of the use of Large Language Models (LLMs) in Appendix H.

## REPRODUCIBILITY STATEMENT

**Theoretical Reproducibility.** For our theoretical part, the setting is clearly stated in Section 3.2, and we also provide a more technical version, including necessary facts and lemmas, in Appendix B. The only assumption of this paper is Assumption 5.1, and two vital conditions is Definition 5.2 and Definition 5.4.

For all Lemmas/Theorems/Corollaries in the main paper, we have:

- Proof sketch of Lemma 4.1 is provided under itself. The formal version and full proof is in Lemma C.6.
- Proof sketch of Lemma 5.3 is provided under itself. The formal version and full proof is in Lemma E.1.
- Proof sketch of Theorem 5.5 is provided under itself. The formal version and full proof is in Theorem F.2.
- Proof sketch of Corollary 5.6 is provided under itself. The formal version and full proof is in Corollary F.3.
- Proof sketch of Theorem 6.1 is provided under itself. The formal version and full proof is in Theorem G.4.
- Proof sketch of Theorem 6.2 is provided under itself. The formal version and full proof is in Theorem G.5.

**Experimental Reproducibility.** We provide the code for the experiments in Section 7 and a markdown file "README.md" for explaining how to reproduce our experimental results.

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

# Appendix

CONTENTS

# A   FORMAL ANALYSIS OF THREE-STAGE GENERALIZATION

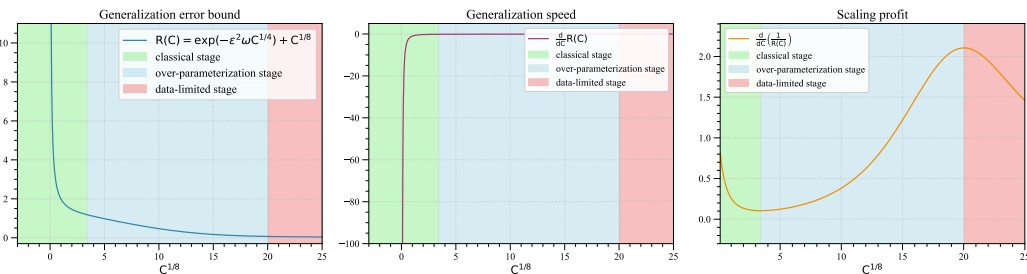

Figure 4: Visualization of Generalization error bound, Generalization speed, and Scaling profit. We use three different colors to distinguish three scaling stages.

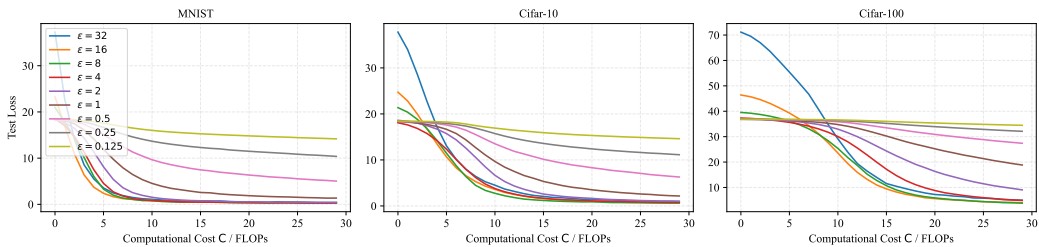

Figure 5: Results of the experiment in Section 7: Test loss curves on MNIST, Cifar-10, 100 datasets with different computational cost $C$ and grokking coefficient $\varepsilon$.

We list the key concepts for our scaling law analysis as follows:

- The generalization error bound, denoted by $R(C)$, is a function of the computational cost $C$. Theorem 6.1 proves that $R(C)$ converges as $C$ increases. This bound is visualized in the left panel of Figure 4.

- The generalization speed, defined as the derivative $R'(C) = \frac{d}{dC}R(C)$, measures the rate of convergence of the generalization error bound. The sign of $R'(C)$ indicates whether the bound is decreasing (converging) or increasing, and its magnitude represents the convergence rate. This is visualized in the middle panel of Figure 4.

- The scaling profit is defined as the derivative of the inverse of the generalization error bound, i.e., $\frac{d}{dC}\left(\frac{1}{R(C)}\right)$. A higher value of this metric indicates a greater benefit (profit) from increasing the computational cost, and conversely, a lower value suggests diminishing returns. This is shown in the right panel of Figure 4.

For the left and middle images in Figure 4, generalization error and speed decrease monotonically to zero as computational cost is scaled up. The right image displays a more complex scaling profit curve, where the computational cost range is divided into three phases by the curve's maximum and minimum points.

The upper bound during the *classical stage* is principally determined by the term $O(\alpha^2/(\omega C^{1/8}))$. The scarcity of neurons initially facilitates a phase of rapid convergence by constraining the model's capacity to fit the data perfectly. This phase is transient, however, as the convergence rate is soon curtailed by the escalating complexity of the model.

The *over-parameterization stage* is marked by a transition in the scaling law. Here, the dominant term in the bound of Theorem 6.1 simplifies to $\exp(-\varepsilon^2 \omega C^{\frac{1}{4}})$, indicating that the risk decreases exponentially with compute (model size and training duration). This stage corresponds to a stable, low-variance solution where bias dominates. We identify this phase with the grokking phenomenon

(Power et al., 2022). Our analysis shows that grokking is expected for any $\varepsilon < \max_{\mathsf{C}>0} \mathcal{T}(\mathsf{C})$, a relationship illustrated in Figure 1.b.

The scaling behavior enters a *data-limited stage* when the number of samples becomes the primary constraint. In this regime, variance is the dominant factor because models can easily memorize the small training set but fail to generalize. The scaling bound for this phase is therefore $O(\alpha^2/(\omega C^{1/8}))$.

## B  TECHNICAL PRELIMINARY

### B.1  NOTATIONS

In this paper, we use integer $d$ to denote the dimension of networks. We use $L$ to denote the input length in language models. $\nabla_x f(x)$ and $\frac{\mathrm{d}f(x)}{\mathrm{d}x}$ are both means to take the derivative of $f(x)$ with $x$. Let a vector $z \in \mathbb{R}^n$. We denote the $\ell_2$ norm as $\|z\|_2 := (\sum_{i=1}^n z_i^2)^{1/2}$, the $\ell_1$ norm as $\|z\|_1 := \sum_{i=1}^n |z_i|$, $\|z\|_0$ as the number of non-zero entries in $z$, $\|z\|_\infty$ as $\max_{i \in [n]} |z_i|$. We use $z^\top$ to denote the transpose of a $z$. We use $\langle \cdot, \cdot \rangle$ to denote the inner product. Let $A \in \mathbb{R}^{n \times d}$, we use $\mathrm{vec}(A)$ to denote a length $nd$ vector. We denote the Frobenius norm as $\|A\|_F := (\sum_{i \in [n], j \in [d]} A_{i,j}^2)^{1/2}$. For any positive integer $n$, we use $[n]$ to denote set $\{1, 2, \cdots, n\}$. We use $\mathbb{E}[]$ to denote the expectation. We use $\Pr[]$ to denote the probability. We use $\epsilon$ to denote the error. We define $\lambda_{\min}(\cdot)$ as a function that outputs the minimum eigenvalues of the input matrix, e.g. matrix $A \in \mathbb{R}^{n \times n}$ has eigenvalues $\{\lambda_1, \lambda_2, \cdots, \lambda_n\}$, $\lambda_{\min}(A) = \min\{\lambda_1, \lambda_2, \cdots, \lambda_n\}$. For a vector $a \in \mathbb{R}^n$, we use $a_i$ to denote its $i$-th entry for $i \in [n]$. For a matrix $A \in \mathbb{R}^{n \times d}$, vector $A_i \in \mathbb{R}^d$ is the $i$-th row for $i \in [n]$ and vector $A_{*,j} \in \mathbb{R}^n$ is the $j$-th column for $j \in [d]$. For a function $f :\to \mathbb{R}^n$, we use $f_i$ to denote the $i$-th entry of its output. For a function $f :\to \mathbb{R}^{n \times d}$, we use $f_i \in \mathbb{R}^d$ to denote the $i$-th row of its output for $i \in [n]$ and us $f_{*,j} \in \mathbb{R}^n$ to denote the $j$-th column of its output for $j \in [d]$. We use $\mathbb{I}\{\mathsf{E}_1, \mathsf{E}_2, \cdots, \mathsf{E}_n\}$ to denote the indicator for event set $\{\mathsf{E}_1, \mathsf{E}_2, \cdots, \mathsf{E}_n\}$, only when $\mathsf{E}_1, \mathsf{E}_2, \cdots, \mathsf{E}_n$ are all true, $\mathbb{I}\{\mathsf{E}_1, \mathsf{E}_2, \cdots, \mathsf{E}_n\} = 1$; otherwise, it equals to 0. For a vector $a \in \mathbb{R}^{d^2}$, function $\mathrm{mat}$ reshapes $a$ to a $d \times d$ matrix, where its $(i,j)$-th entry is $\mathrm{mat}_{i,j}(a) = a_{(i-1)d+j}$ for $(i,j) \in [d] \times [d]$. For a matrix $A \in \mathbb{R}^{n \times d}$, function $\mathrm{vec}$ flattens $A$ to a $nd$-dimensional vector, where its $i$-th entry is $\mathrm{vec}_i(A) = A_{\lfloor i/d \rfloor, i - \lfloor i/d \rfloor \cdot d}$ for $i \in [nd]$.

### B.2  ORIGINAL MODEL DEFINITIONS

**Definition B.1** (Weights and Initialization). *The weights of the model are denoted as $\theta(t)$ where $t \geq 0$ is time. It contains the weights of each layer $\theta(t) = \{\theta_{(\nu)}(t)\}_{\nu=1}^N$, and $\theta_{(\nu)} = \{U_{(\nu)}(t), W_{(\nu)}(t), A_{(\nu)}\}$. For each matrix:*

- *Each entry of $U_{(\nu)}(0) \in \mathbb{R}^{d \times d}$ is initialized from the standard Gaussian distribution, formally, $U_{(\nu),k_1,k_2}(0) \sim \mathcal{N}(0,1)$ for any $k_1, k_2 \in [d]$.*

- *Each entry of $W_{(\nu)}(0) \in \mathbb{R}^{d \times m}$ is initialized from the standard Gaussian distribution, formally, $w_{(\nu),r,k}(0) \sim \mathcal{N}(0,1)$ for any $r \in [m], k \in [d]$, where $w_{(\nu),r}(0) \in \mathbb{R}^d$ is the $r$-th column of $W_{(\nu)}(0)$.*

- *Each entry of $A_{(\nu)}(0) \in \mathbb{R}^{m \times d}$ is initialized from a $\pm 1$ uniform distribution, formally, $a_{(\nu),r,k}(0) \sim \mathsf{Uniform}\{-1,+1\}$ for any $r \in [m], k \in [d]$, where $a_{(\nu),r}(0) \in \mathbb{R}^d$ is the $r$-th row of $A_{(\nu)}(0)$.*

**Definition B.2** (Data Distribution). *We denote $F^* : \mathcal{X} \to [C_1, C_2]^{L \times d}$ as the target function. The $\mathcal{X} \in \mathbb{R}^{L \times d}$ is a combination of $L$ $d$-dimensional balls where $\|X_\ell\|_2^2 = \Theta(1), \forall X \in \mathcal{X}, \ell \in [L]$. $C_1, C_2 \in \mathbb{R}$ are two fixed constants. The distribution: $\mathcal{D} = \{(X, F^*(X) + \Xi), \Xi \in \mathbb{R}^{L \times d}$ is some random noise$\} \subset \mathcal{X} \times \mathcal{Y}$. The random noise $\Xi \in \mathbb{R}^{L \times d}$ is centered by $\mathbf{0}_{L \times d}$.*

**Definition B.3** (Original Dataset). *The original dataset is $\mathbb{D} = \{(X_i, Y_i)\}_{i=1}^n \subset \mathcal{D}$, where $X_i, Y_i = F^*(X_i) + \Xi_i \in \mathbb{R}^{L \times d}$. For each data point $(X_i, Y_i), \forall i \in [n]$, it holds that:*

- $\|X_{i,\ell}\|_2 = \Theta(1)$ *for $\ell \in [L]$.*

**Definition B.4** (Model Functions). *Given an input matrix $X \in \mathbb{R}^{L \times d}$, the model function is given by ($\varepsilon > 0$ is the grokking coefficient):*

$$F(X, \theta(t)) := \varepsilon \cdot F_{(N)}(F_{(N-1)}(\cdots F_{(2)}(F_{(1)}(X + E, \theta(t)), \theta(t)) \cdots), \theta(t)).$$

*We list the original definition of each function as follows:*

- 

$$F_{(\nu)}(X, \theta(t)) := \frac{\omega}{\sqrt{m}} \mathrm{ReLU}\left(\mathrm{Softmax}\left(\kappa \cdot XU_{(\nu)}(t)X^\top + M\right) XW_{(\nu)}(t)\right) A_{(\nu)}$$

- $\mathrm{Softmax}\,(A) := \mathrm{diag}(\exp(A)\mathbf{1}_L)^{-1} \cdot \exp(A) \in \mathbb{R}^{L \times L}$ *for* $A \in \mathbb{R}^{L \times L}$.

- $\mathrm{ReLU}_{\ell,k}(X) := \max\{X_{\ell,k}, 0\}$ *for any* $\ell \in [L]$, $k \in [d]$, $X \in \mathbb{R}^{L \times d}$.

- $M_{\ell_1,\ell_2} := \begin{cases} 0, & \ell_1 \geq \ell_2 \\ -\infty. & \ell_1 < \ell_2 \end{cases}, \forall \ell_1, \ell_2 \in [L]$.

**Definition B.5.** *We denote the special notations:*

- *We denote* $w_{(\nu),r}(t) \in \mathbb{R}^d$ *as the* $r$-*th column of* $W_{(\nu)} \in \mathbb{R}^{d \times m}$ *for* $r \in [m]$, $\nu \in [N]$.

- *We denote* $a_{(\nu),r} \in \mathbb{R}^d$ *as the* $r$-*th row of* $A_{(\nu)} \in \mathbb{R}^{m \times d}$ *for* $r \in [m]$, $\nu \in [N]$.

**Definition B.6.** *We define the training objective:*

$$\mathcal{L}(t, \mathbb{D}) := \mathop{\mathbb{E}}_{(X,Y)\sim\mathbb{D}}[\|F(X, \theta(t)) - Y\|_F^2]$$

### B.3 BASIC FACTS AND LEMMAS

**Fact B.7.** *For a variable* $x \sim \mathcal{N}(0, \sigma^2)$, *then with probability at least* $1 - \delta$, *we have:*

$$|x| \leq C\sigma\sqrt{\log(1/\delta)}$$

**Fact B.8.** *For an* 1-*Lipschitz function* $f(\cdot)$, *we have:*

$$|f(x) - f(y)| \leq |x - y|, \forall x, y \in \mathbb{R}^d$$

**Fact B.9.** *For a Gaussian variable* $x \sim \mathcal{N}(0, \sigma^2 \cdot I_d)$ *where* $\sigma \in \mathbb{R}$, *then for any* $t > 0$, *we have:*

$$\Pr[x \leq t] \leq \frac{2t}{\sqrt{2\pi}\sigma}$$

**Fact B.10.** *For a Gaussian vector* $w \sim \mathcal{N}(0, \sigma^2 \cdot I_d)$ *where* $\sigma \in \mathbb{R}$, *and a fixed vector* $x \in \mathbb{R}^d$, *we have:*

$$w^\top x \sim \mathcal{N}(0, \sigma^2 \|x\|_2^2)$$

**Fact B.11.** *For two matrices* $H, \widetilde{H} \in \mathbb{R}^{n \times n}$, *we have:*

$$\lambda_{\min}(\widetilde{H}) \geq \lambda_{\min}(H) - \|H - \widetilde{H}\|_F$$

**Fact B.12.** *The Lipchitz constant of the softmax function is bounded by* $O(1)$, *such that:*

$$\|\langle\exp(x), \mathbf{1}_L\rangle^{-1}\exp(x) - \langle\exp(y), \mathbf{1}_L\rangle^{-1}\exp(y)\|_2 \leq O(1) \cdot \|x - y\|_2, \forall x, y \in \mathbb{R}^L.$$

**Fact B.13.** *For a matrix* $H \in \mathbb{R}^{n \times n}$, *there is* $\lambda_{\min}(H \otimes I_d) = \lambda_{\min}(H)$.

In addition, we state four vital lemmas of concentration inequalities for simplifying analysis:

**Lemma B.14** (Hoeffding bound). *Let* $X_1, \cdots, X_n$ *denote* $n$ *independent bounded variables in* $[a_i, b_i]$ *for* $a_i, b_i \in \mathbb{R}$. *Let* $X := \sum_{i=1}^n X_i$, *then we have*

$$\Pr[|X - \mathbb{E}[X]| \geq t] \leq 2\exp(-\frac{2t^2}{\sum_{i=1}^n (b_i - a_i)^2})$$

**Lemma B.15** (Markov's inequality). *If* $X$ *is a non-negative random variable and* $a > 0$, *then the probability that* $X$ *is at least* $a$ *is at most the expectation of* $X$ *divided by* $a$:

$$\Pr[X \geq a] \leq \frac{\mathbb{E}[X]}{a}$$

**Lemma B.16** (Chernoff bound). *Let* $X = \sum_{i=1}^n X_i$, *where* $X_i = 1$ *with probability* $p_i$ *and* $X_i = 0$ *with probability* $1 - p_i$, *and all* $X_i$ *are independent. Let* $\mu = \mathbb{E}[X] = \sum_{i=1}^n p_i$. *Then*

- $\Pr[X \geq (1 + \delta)\mu] \leq \exp(-\delta^2\mu/3)$, $\forall \delta > 0$;

- $\Pr[X \leq (1 - \delta)\mu] \leq \exp(-\delta^2\mu/1)$, $\forall 0 < \delta < 1$.

## C  GRADIENT COMPUTATION AND LEARNING DYNAMICS

### C.1  SIMPLIFIED MODEL DEFINITIONS

**Definition C.1** (Rearranged Dataset). *Given the origin dataset $\mathbb{D} = \{X_i, Y_i\}_{i=1}^n \subset \mathbb{R}^{L \times d} \times \mathbb{R}^{L \times d}$. The rearranged dataset is $\mathbb{D}_{\mathrm{rearrange}} = \{(X_{i,\leq \ell} + E_{\leq \ell}, Y_{i,\ell})\}_{(i,\ell)=(1,1)}^{(n,L)}$, where $X_{i,\leq \ell} + E_{\leq \ell} \in \mathbb{R}^{\ell \times d}$ and $Y_{i,\ell} \in \mathbb{R}^d$.*

**Definition C.2** (Simplified Model Function). *Given the origin dataset $\mathbb{D} = \{X_i, Y_i\}_{i=1}^n \subset \mathbb{R}^{L \times d} \times \mathbb{R}^{L \times d}$, we define the compact form of the model function as:*

$\mathsf{F}(t) \in \mathbb{R}^{nL \times d}$, *where its $i$-th row is given by* $\mathsf{F}_p(t) := F_\ell(X_i, \theta(t)) \in \mathbb{R}^d, p \in [nL]$,

$\mathsf{Y} \in \mathbb{R}^{nL \times d}$, *where its $i$-th row is given by* $\mathsf{Y}_p := Y_{i,\ell} \in \mathbb{R}^d, p \in [nL]$.

*Here, $i = \lfloor p/L \rfloor$ and $\ell = p \bmod L$.*

*We list the notation-simplified definitions of all functions as follows:*

- *(Hidden State)* $\Lambda_{(\nu),i}(t) := F_{(\nu)}(\Lambda_{(\nu-1),i}(t), \theta(t)) \in \mathbb{R}^{L \times d}$ *for* $\nu \in [N]$, $\Lambda_{(0),i}(t) = X_i + E$.

- *(Attention Scores)* $\sigma_{(\nu),(i-1)L+\ell}(X) = \mathrm{Softmax}_\ell(\Lambda_{(\nu),i}(t)U_{(\nu)}(t)\Lambda_{(\nu),i}(t)^\top + M) \in \mathbb{R}^L$.

- *(Attention Output)* $o_{(\nu),(i-1)L+\ell}(t) := \Lambda_{(\nu-1),i}(t)^\top \cdot \sigma_{(\nu),(i-1)L+\ell}(t) \in \mathbb{R}^d$.

- *($\ell$-th Token of Hidden State)* $\mu_{(\nu),(i-1)L+\ell}(t) := \frac{\omega}{\sqrt{m}} \sum_{r=1}^m a_{(\nu),r} \cdot \phi(\langle o_{(\nu),(i-1)L+\ell}(t), w_{(\nu),r}(t) \rangle) \in \mathbb{R}^d$, *where* $\phi(x) := \max\{0, x\}, \forall x \in \mathbb{R}$. $\mu_{(0),(i-1)L+\ell}(t) = X_{i,\ell} + E_\ell$.

- *(Model Output)* $\mathsf{F}_{(i-1)L+\ell}(t) = \varepsilon \cdot \sum_{\nu=0}^N \mu_{(\nu),(i-1)L+\ell}(t) \in \mathbb{R}^d$.

**Lemma C.3.** *We have:*

$$\mathcal{L}(t, \mathbb{D}) = \frac{1}{n}\|\mathsf{F}(t) - \mathsf{Y}\|_F^2$$

*Proof.* Since the decoder-only property of the model function, we have:

$$F_{(i-1)L+\ell}(t) = F_\ell(X_i, \theta(t)), \forall (X_i, Y_i) \in \mathbb{D}, i \in [n], \ell \in [L].$$

We then separate each token vector with:

$$\mathcal{L}(t, \mathbb{D}) = \mathop{\mathbb{E}}_{(X,Y) \sim \mathbb{D}} [\|F(X, \theta(t)) - Y\|_F^2]$$

$$= \frac{1}{n} \sum_{(X,Y) \in \mathbb{D}} \|F(X, \theta(t)) - Y\|_F^2$$

$$= \frac{1}{n} \sum_{(X,Y) \in \mathbb{D}} \sum_{\ell=1}^L \|F_\ell(X, \theta(t)) - Y_\ell\|_2^2$$

$$= \frac{1}{n} \|\mathsf{F}(t) - \mathsf{Y}\|_F^2,$$

where the first three steps follow from simple algebra, and the last step follows from the definitions of $\mathsf{F}$ and $\mathsf{Y}$. $\square$

### C.2  GRADIENT COMPUTATION

**Lemma C.4.** *For $\nu \in [N]$, we have:*

- **Part 1.** *We have:*

$$\frac{\mathrm{d}\mathcal{L}(t, \mathbb{D})}{\mathrm{d}\,\mathrm{vec}(U_{(\nu)}(t))} = \frac{\omega \cdot \kappa}{\sqrt{m}} \sum_{p=1}^{nL} (\Lambda_{(\nu-1),i,\ell,*}(t) \otimes \Lambda_{(\nu-1),i}(t))^\top \left(\mathrm{diag}(\sigma_{(\nu),p}(t)) - \sigma_{(\nu),p}(t)\sigma_{(\nu),p}(t)^\top\right)$$

$$\Lambda_{(\nu-1),i}(t) \sum_{r \in [m]} \langle \frac{\mathrm{d}\mathcal{L}(t,\mathbb{D})}{\mathrm{d}\mu_{(\nu),p}(t)}, a_{(\nu),r} \rangle \cdot w_{(\nu),r}(t)\, \mathbb{I}\{o_{(\nu),p}(t)^\top w_{(\nu),r}(t) > 0\},$$

*where $i = \lfloor p/L \rfloor$ and $\ell = p \bmod L$.*

- **Part 2.** *For any $r \in [m]$, we have:*

$$\frac{\mathrm{d}\mathcal{L}(t,\mathbb{D})}{\mathrm{d}w_{(\nu),r}(t)} = \frac{\omega}{\sqrt{m}} \sum_{p=1}^{nL} \langle \frac{\mathrm{d}\mathcal{L}(t,\mathbb{D})}{\mathrm{d}\mu_{(\nu),p}(t)}, a_{(\nu),r} \rangle \cdot o_{(\nu),p}(t) \cdot \mathbb{I}\{w_{(\nu),r}(t)^\top o_{(\nu),p}(t) > 0\}.$$

- **Part 3.** *For $\nu \in [N]$ and $p \in [nL]$, we have:*

$$\frac{\mathrm{d}\mathcal{L}(t,\mathbb{D})}{\mathrm{d}\mu_{(\nu),p}(t)} = \frac{2\varepsilon}{n} \cdot \left(I_d + \mathrm{diag}(G_{(\nu),p}(t))\right)(\mathsf{F}_p(t) - \mathsf{Y}_p),$$

*where*

$$G_{(\nu),p}(t) = \frac{\omega}{\sqrt{m}} \Big( \sigma_{(\nu),p,\ell}(t) \cdot I_d$$
$$+ \kappa U_{(\nu)}(t) \Lambda_{(\nu-1),i}(t)^\top (\mathrm{diag}(\mathbf{1}_n - \frac{1}{2}e_\ell)) \cdot \big(\mathrm{diag}(\sigma_{(\nu),p}(t)) - \sigma_{(\nu),p}(t)\sigma_{(\nu),p}(t)^\top\big) \Lambda_{(\nu-1),i}(t)\Big)$$
$$\cdot \sum_{r \in [m]} \langle \frac{\mathrm{d}\mathcal{L}(t,\mathbb{D})}{\mathrm{d}\mu_{(\nu+1),p}(t)}, a_{(\nu),r} \rangle \cdot w_{(\nu),r}(t)\, \mathbb{I}\{o_{(\nu),p}(t)^\top w_{(\nu),r}(t) > 0\}),$$

*and $G_{(N),p}(t) = \mathbf{0}_d$.*

*Here, we denote $i = \lfloor p/L \rfloor$ and $\ell = p \bmod L$. $e_\ell$ is a $n$-dimensional one-hot vector with the $r$-th entry equal $1$.*

*Proof.* We omit the lemma proof since the gradient computation trivially follows from basic algebra and multivariable chain rules. $\square$

### C.3 Learning Dynamics

We give the definition of NTK:

**Definition C.5.** *We define the kernel matrix at $\nu$-th layer as $H_{(\nu)} \in \mathbb{R}^{nL \times nL}$ and its $(i,j)$-th entry ($\forall (p,q) \in [nL] \times [nL]$) is defined as:*

$$H_{(\nu),p,q}(t) := \underbrace{\langle \beta_{(\nu),p}(t), \beta_{(\nu),q}(t) \rangle}_{\text{kernel w.r.t. } W_{(\nu)}(t)} + \underbrace{\langle \gamma_{(\nu),p}(t), \gamma_{(\nu),q}(t) \rangle}_{\text{kernel w.r.t. } U_{(\nu)}(t)},$$

*Here, we let:*

$$\beta_{(\nu),p}(t) := \frac{\omega}{\sqrt{m}} \underbrace{o_{(\nu),p}(t)}_{d \times 1} \otimes \underbrace{\mathbf{1}_{W_{(\nu)}(t)^\top o_{(\nu),p}(t) > 0}}_{m \times 1} \in \mathbb{R}^{md},$$

$$\gamma_{(\nu),p}(t) := \frac{\omega \cdot \kappa}{\sqrt{m}} \underbrace{(\Lambda_{(\nu-1),i,\ell,*}(t) \otimes \Lambda_{(\nu-1),i}(t))^\top}_{d^2 \times L} \underbrace{\big(\mathrm{diag}(\sigma_{(\nu),p}(t)) - \sigma_{(\nu),p}(t)\sigma_{(\nu),p}(t)^\top\big)}_{L \times L}$$
$$\underbrace{\Lambda_{(\nu-1),i}(t)}_{L \times d} \sum_{r \in [m]} \underbrace{w_{(\nu),r}(t)\, \mathbb{I}\{o_{(\nu),p}(t)^\top w_{(\nu),r}(t) > 0\}}_{d \times 1} \in \mathbb{R}^{d^2},$$

*where $\otimes$ is the Kronecker product and $i = \lfloor p/L \rfloor$, $\ell = p \bmod L$. The indicator vector $\mathbf{1}_{W_{(\nu)}(t)^\top o_{(\nu),p}(t) > 0} \in \{0,1\}^m$ where its $r$-th entry is $\mathbb{I}\{(W_{(\nu)}(t)^\top o_{(\nu),p}(t))_r > 0\}$ for $r \in [m]$.*

We restate Lemma 4.1 below as its formal version:

**Lemma C.6** (Formal version of Lemma 4.1). *The learning dynamics of the multi-layer transformer Eq. (1) is given by:*

$$\mathbb{E}[\frac{\mathrm{d}}{\mathrm{d}t}\mathcal{L}(t,\mathbb{D})] = - \sum_{\nu\in[N]} \underbrace{\mathrm{vec}\left(\frac{\mathrm{d}}{\mathrm{d}\mu_{(\nu)}(t)}\mathcal{L}(t,\mathbb{D})\right)^{\top}}_{1\times nLd} \cdot \underbrace{\left(H_{(\nu)}(t)\otimes I_d\right)}_{nLd\times nLd} \cdot \underbrace{\mathrm{vec}\left(\frac{\mathrm{d}}{\mathrm{d}\mu_{(\nu)}(t)}\mathcal{L}(t,\mathbb{D})\right)}_{nLd\times 1}$$

*where $\mu_{(\nu)}(t)$ is a $nL \times d$ matrix, $\mu_{(\nu),p}(t)$ is the $(p \bmod L)$-th row of $v$-th layer output regarding to input matrix $X_{\lfloor p/L\rfloor}$ for any $p \in [nL]$ and $\nu \in [N]$.*

*Proof.* We have:

$$\mathbb{E}[\frac{\mathrm{d}}{\mathrm{d}t}\mathcal{L}(t,\mathbb{D})] = \mathbb{E}[\sum_{\nu=1}^{N}(\frac{\mathrm{d}\mathcal{L}(t,\mathbb{D})}{\mathrm{d}\,\mathrm{vec}(W_{(\nu)}(t))}^{\top}\frac{\mathrm{d}\,\mathrm{vec}(W_{(\nu)}(t))}{\mathrm{d}t} + \frac{\mathrm{d}\mathcal{L}(t,\mathbb{D})}{\mathrm{d}\,\mathrm{vec}(U_{(\nu)}(t))}^{\top}\frac{\mathrm{d}\,\mathrm{vec}(U_{(\nu)}(t))}{\mathrm{d}t})]$$

$$= -\sum_{\nu=1}^{N}(\frac{\mathrm{d}\mathcal{L}(t,\mathbb{D})}{\mathrm{d}\,\mathrm{vec}(W_{(\nu)}(t))}^{\top}\frac{\mathrm{d}\mathcal{L}(t,\mathbb{D})}{\mathrm{d}\,\mathrm{vec}(W_{(\nu)}(t))} + \frac{\mathrm{d}\mathcal{L}(t,\mathbb{D})}{\mathrm{d}\,\mathrm{vec}(U_{(\nu)}(t))}^{\top}\frac{\mathrm{d}\mathcal{L}(t,\mathbb{D})}{\mathrm{d}\,\mathrm{vec}(U_{(\nu)}(t))}$$

$$= -\sum_{\nu=1}^{N}\sum_{p=1}^{nL}\sum_{q=1}^{nL}\frac{\mathrm{d}\mathcal{L}(t,\mathbb{D})}{\mathrm{d}\mu_{(\nu),p}(t)}^{\top}\left(\langle\beta_{(\nu),p}(t),\beta_{(\nu),q}(t)\rangle + \langle\gamma_{(\nu),p}(t),\gamma_{(\nu),q}(t)\rangle\right)\cdot I_d\frac{\mathrm{d}\mathcal{L}(t,\mathbb{D})}{\mathrm{d}\mu_{(\nu),q}(t)}$$

$$= -\sum_{\nu\in[N]}\mathrm{vec}\left(\frac{\mathrm{d}}{\mathrm{d}\mu_{(\nu)}(t)}\mathcal{L}(t,\mathbb{D})\right)^{\top}\cdot\left(H_{(\nu)}(t)\otimes I_d\right)\cdot\mathrm{vec}\left(\frac{\mathrm{d}}{\mathrm{d}\mu_{(\nu)}(t)}\mathcal{L}(t,\mathbb{D})\right),$$

where the first step follows from the chain rule, the second step follows from the gradient flow (Eq. (2)), the third step follows from Part 1, Part 2 and Part 3 of Lemma C.4 and the definitions of $\beta_{(\nu),p}(t)$ and $\gamma_{(\nu),p}(t)$, the last step follows from the definition of $H_{(\nu)}(t)$. □

# D TOOLKIT: HELPFUL BOUNDARIES

We give the lemma about all help bounds in the range of this paper as follows:

**Lemma D.1.** *Denote failure probability* $\delta \in (0, 0.1)$. *Define* $B := \max\{O(\sqrt{\log(Lmd/\delta)}), 1\}$. *Assuming there exists a constant* $R \in (0, 1)$ *satisfying* $\|w_{(\nu),r}(t) - w_{(\nu),r}(0)\|_2 \leq R$ *and* $\|U_{(\nu)}(t) - U_{(\nu)}(0)\|_F \leq R$ *for* $r \in [m]$ *and* $\nu \in [N]$ *and* $R \in (0, 1)$.

*We mark the indices as:* $r \in [m]$, $p \in [nL]$, $\ell_1, \ell_2 \in [L]$, *and* $\nu \in [N]$. *We denote* $i = \lfloor p/L \rfloor$, $\ell = p \bmod L$, $\ell' \in [\ell]$.

*If Definition 5.2 holds, then with a probability at least* $1 - \delta$, *we have:*

- Basic Bounds.

    - **Part 1.** $\|w_{(\nu),r}(0)\|_2 \leq O(\sqrt{d}B)$.
    - **Part 2.** $\|U_{(\nu)}(0)\|_F \leq O(dB)$.
    - **Part 3.** $\|w_{(\nu),r}(t)\|_2 \leq O(\sqrt{d}B)$.
    - **Part 4.** $\|U_{(\nu)}(t)\|_F \leq O(dB)$.
    - **Part 5.** $\|\Lambda_{(\nu),i,\ell_1}(t)\|_2 = \Theta(1)$.
    - **Part 6.** $\|\Lambda_{(\nu),i}(t)U_{(\nu)}(t)\Lambda_{(\nu),i}(t)^\top\|_\infty \leq O(dB)$.
    - **Part 7.** $\exp_{\ell_1,\ell_2}(\Lambda_{(\nu),i}(t)U_{(\nu)}(t)\Lambda_{(\nu),i}(t)^\top) \in [\exp(-O(dB), \exp(O(dB))]$.
    - **Part 8.** *For* $\ell' \in [\ell]$, $\sigma_{(\nu),p,\ell'}(t) \in [\exp(-O(dB))/L, 1]$.

- Perturbation Bounds.

    - **Part 9.** $\|w_{(\nu),r}(t) - w_{(\nu),r}(0)\|_2 \leq R$.
    - **Part 10.** $\|U_{(\nu)}(t) - U_{(\nu)}(0)\|_F \leq R$.
    - **Part 11.** $\|\Lambda_{(\nu),i,\ell}(t) - \Lambda_{(\nu),i,\ell}(0)\|_2 \leq O(R)$.
    - **Part 12.** $\|\sigma_{(\nu),p}(t) - \sigma_{(\nu),p}(0)\|_2 \leq O(\sqrt{L}R)$.
    - **Part 13.** $\|o_{(\nu),p}(t) - o_{(\nu),p}(0)\|_2 \leq O(\sqrt{L}R)$.

- Gradient and Function Norms.

    - **Part 14.** $\mathcal{L}(t, \mathbb{D}) \leq O(Ld)$.
    - **Part 15.** $\|\frac{d\mathcal{L}(t,\mathbb{D})}{d\mu_{(\nu)}(t)}\|_F^2 \asymp \varepsilon^2 \cdot \mathcal{L}(t, \mathbb{D})$.
    - **Part 16.** $\|\gamma_{(\nu),p}(t)\|_2 \leq o(1/\sqrt{m})$.

*Proof.* **Proof of Part 1.** This proof follows from the initialization of $W_{(\nu)}(0)$ and the union bound of the tail bound of the Gaussian distribution (Fact B.7).

**Proof of Part 2.** This proof follows from initialization of $U_{(\nu)}(0)$ and the union bound of the tail bound of the Gaussian distribution (Fact B.7).

**Proof of Part 3.** This proof combines $\|w_{(\nu),r}(t) - w_{(\nu),r}(0)\|_2 \leq R$, triangle inequality and $R \leq B$.

**Proof of Part 4.** This proof combines $\|U_{(\nu)}(t) - U_{(\nu)}(0)\|_F \leq R$, triangle inequality and $R \leq B$.

**Proof of Part 5.** We have:

$$\|\Lambda_{(0),i,\ell}(t)\|_2 = \Theta(1).$$

Therefore, we have:

$$\|o_{(1),p}(t)\|_2 = \|\Lambda_{(0),i}(t)^\top \sigma_{(\nu),p}(t)\|_2 = \Theta(1),$$

where this step follows from the property that the sum of any softmax vector is $1$.

Therefore, we can show that:

$$\|\mu_{(\nu),p}(t)\|_2 = \|\frac{\omega}{\sqrt{m}} \sum_{r=1}^m a_{(1),r} \cdot \phi(\langle o_{(1),p}(t), w_{(1),r}(t)\rangle)\|_2$$

$$\leq \frac{\omega\sqrt{d}}{\sqrt{m}} \| \sum_{r=1}^{m} a_{(1),r} \cdot \phi(\langle o_{(1),p}(t), w_{(1),r}(t) \rangle) \|_{\infty}$$

$$\leq \frac{\omega\sqrt{d}}{\sqrt{m}} \max_{k \in [d]} | \sum_{r=1}^{m} a_{(1),r,k} \cdot \phi(\langle o_{(1),p}(t), w_{(1),r}(t) \rangle) |$$

where the first step follows from the definition of $\mu_{(\nu),p}(t)$, the second step follows from simple algebras, the third step follows from the definition of infinite norm.

We apply Hoeffding bound to each variable $a_{(1),r,k} \cdot \phi(\langle o_{(1),p}(t), w_{(1),r}(t) \rangle)$, we have:

$$|a_{(1),r,k} \cdot \phi(\langle o_{(1),p}(t), w_{(1),r}(t) \rangle)| \leq O(\sqrt{d}B),$$
$$\mathbb{E}[a_{(1),r,k} \cdot \phi(\langle o_{(1),p}(t), w_{(1),r}(t) \rangle)] = 0.$$

With a probability at least $1 - \delta$, we have:

$$\|\mu_{(\nu),p}(t)\|_2 \leq \frac{\omega\sqrt{d}}{\sqrt{m}} \max_{k \in [d]} | \sum_{r=1}^{m} a_{(1),r,k} \cdot \phi(\langle o_{(1),p}(t), w_{(1),r}(t) \rangle) |$$

$$\leq \frac{\omega\sqrt{d}}{\sqrt{m}} \cdot O(\sqrt{m}dB)\sqrt{\log(m/\delta)} \leq O(\omega dB^2).$$

Hence,

$$\|\Lambda_{(1),i,\ell}(t)\|_2 = \| \frac{\omega}{\sqrt{m}} \sum_{r=1}^{m} a_{(1),r} \cdot \phi(\langle o_{(1),p}(t), w_{(1),r}(t) \rangle) + \Lambda_{(0),i,\ell}(t) \|_2$$

$$= \Theta(1) \pm O(\omega dB^2)$$

$$= \Theta(1) \pm o(\frac{1}{N})$$

the last step follows from choosing $\omega = o(\frac{1}{NdB^2})$.

By induction, we can get:

$$\|\Lambda_{(\nu),i,\ell}(t)\|_2 = \Theta(1),$$

and

$$\|o_{(\nu),p}(t)\|_2 = \Theta(1). \tag{4}$$

**Proof of Part 6.** We have:

$$\|\Lambda_{(\nu),i}(t) U_{(\nu)}(t) \Lambda_{(\nu),i}(t)^{\top}\|_{\infty} = \max_{(\ell_1, \ell_2) \in [L] \times [L]} |\Lambda_{(\nu),i,\ell_1}(t)^{\top} U_{(\nu)}(t) \Lambda_{(\nu),i,\ell_2}(t)|$$

$$\leq O(dB)$$

where the first step follows from simple algebra, the second step follows from Part 4 and Part 5 of this lemma and Cauchy-Schwarz inequality.

**Proof of Part 7.** This proof follows from Part 3 of this lemma and simple algebra.

**Proof of Part 8.** Following Part 7 of this lemma, we can show that:

$$\sum_{\ell' \in [\ell]} \exp(\Lambda_{(\nu),i,\ell}(t)^{\top} U_{(\nu)}(t) \Lambda_{(\nu),i,\ell'}(t)) \leq \ell \exp(O(dB)) \leq L \exp(O(dB)).$$

Then we can show that:

$$\sigma_{(\nu),p,\ell'}(t) \geq \exp(\Lambda_{(\nu),i,\ell}(t)^{\top} U_{(\nu)}(t) \Lambda_{(\nu),i,\ell'}(t)) / \sum_{\ell' \in [\ell]} \exp(\Lambda_{(\nu),i,\ell}(t)^{\top} U_{(\nu)}(t) \Lambda_{(\nu),i,\ell'}(t))$$

$$\geq \frac{\exp(-O(dB))}{L \exp(O(dB))}$$

$$\geq \frac{\exp(-O(dB))}{L}.$$

Combining the previous results, we obtain the result of this part.

**Proof of Part 9.** Directly from the lemma condition $\|w_{(\nu),r}(t) - w_{(\nu),r}(0)\|_F \leq R$.

**Proof of Part 10.** Directly from the lemma condition $\|U_{(\nu)}(t) - U_{(\nu)}(0)\|_F \leq R$.

**Proof of Part 11, 12 and 13.** When $\nu = 1$, we have:

$$\|\Lambda_{(1),i,\ell}(t) - \Lambda_{(1),i,\ell}(0)\|_F$$

$$= \sqrt{\sum_{k\in[d]} \left(\frac{\omega}{\sqrt{m}} \cdot \sum_{r=1}^{m} a_{(1),r,k}\left(\phi(\langle w_{(1),r}(t), o_{(1),p}(t)\rangle) - \phi(\langle w_{(1),r}(0), o_{(1),p}(0)\rangle)\right)\right)^2},$$

besides, $\Lambda_{(0),i,\ell}(t) = \Lambda_{(0),i,\ell}(0)$.

Next, we have:

$$\|\sigma_{(1),p}(t) - \sigma_{(1),p}(0)\|_2 \leq O(1) \cdot \|\Lambda_{(0),i}(0)(U_{(1)}(t) - U_{(1)}(0))^\top \Lambda_{(0),i,\ell}(0)\|_2$$
$$\leq O(\sqrt{L}R)$$

where the first step follows from the definition of $\sigma_{(1),i}(t)$ and Fact B.12, the second step follows from $\|U_{(1)}(t) - U_{(1)}(0)\|_F \leq R$, $\|\Lambda_{(0),i}(0)\|_F = O(\sqrt{L})$, $\|\Lambda_{(0),i,\ell}(0)\|_2 = O(1)$ and Cauchy-Schwarz inequality.

Thus, we can show that

$$\|o_{(1),i}(t) - o_{(1),i}(0)\|_2 = \|\Lambda_{(0),i}(0)^\top \sigma_{(1),p}(t) - \Lambda_{(0),i}(0)^\top \sigma_{(1),p}(0)\|_2$$

$$= \sqrt{\sum_{\ell'=1}^{\ell} (\sigma_{(1),p,\ell'}(t) - \sigma_{(1),p,\ell'}(t))^2 \|\Lambda_{(0),i,\ell}(0)\|_2^2}$$

$$\leq \sqrt{\sum_{\ell'=1}^{\ell} (\sigma_{(1),p,\ell'}(t) - \sigma_{(1),p,\ell'}(t))^2 O(1)}$$

$$= O(1) \cdot \|\sigma_{(1),p}(t) - \sigma_{(1),p}(t)\|_2$$

$$\leq O(\sqrt{L}R),$$

where these steps follow from some basic algebra, Fact B.12 and Part 10 of this Lemma.

For a certain index $k \in [d]$, we apply Hoeffding's inequality (Lemma B.14) to each random variable $a_{(1),r} \cdot \phi(\langle w_{(1),t}(t), o_{(1),i}(t)\rangle)$, besides, we have:

$$|a_{(1),r,k} \cdot (\phi(\langle w_{(1),r}(t), o_{(1),i}(t)\rangle) - \phi(\langle w_{(1),r}(0), o_{(1),i}(0)\rangle))|$$
$$\leq \max\{\|\|w_{(1),r}(0)\| \cdot O(\sqrt{L}R), \|o_{(1),i}(t)\|_2 \cdot O(R)\}$$
$$= O(\sqrt{L}dBR),$$
$$\mathbb{E}[a_{(1),r,k} \cdot (\phi(\langle w_{(1),r}(t), o_{(1),i}(t)\rangle) - \phi(\langle w_{(1),r}(0), o_{(1),i}(0)\rangle))] = 0.$$

Then with a probability at least $1 - \delta$, we have:

$$\frac{\omega}{\sqrt{m}} \sum_{r=1}^{m} a_{(1),r,k} \cdot \phi(\langle w_{(1),t}(t), o_{(1),i}(t)\rangle) \leq O(\frac{\omega}{\sqrt{m}}BR)\sqrt{Lmd\log(1/\delta)} \leq O(R/(Bd\sqrt{L}d))$$

where the last step follows from choosing $\omega = o(\frac{1}{\sqrt{L}d^2B^3})$.

We therefore get $\|\Lambda_{(1),i,\ell}(t) - \Lambda_{(1),i,\ell}(0)\|_2 \leq O(R)$ for all $\ell \in [\ell]$ and $\|\Lambda_{(1),i,\ell}(t) - \Lambda_{(1),i,\ell}(0)\|_\infty \leq O(R/\sqrt{d})$.

When $\nu = 2$, we have:

$$\|\sigma_{(2),p}(t) - \sigma_{(2),p}(t)\|_2$$

$$\leq O(1) \cdot \|\Lambda_{(1),i}(t) U_{(2)}(t)^\top \Lambda_{(1),i,\ell}(t) - \Lambda_{(1),i}(0) U_{(2)}(0)^\top \Lambda_{(1),i,\ell}(0)\|_2$$

$$= O(1) \cdot \|(\Lambda_{(1),i}(t) \otimes \Lambda_{(1),i,\ell}(t)) \operatorname{vec}(U_{(2)}(t))\rangle - (\Lambda_{(1),i}(t) \otimes \Lambda_{(1),i,\ell}(t)) \operatorname{vec}(U_{(2)}(0))\|_2$$

$$\leq O(1) \cdot \max\{\|\Lambda_{(1),i}(t) \otimes \Lambda_{(1),i,\ell}(t)\|_F \cdot O(R),$$
$$\|U_{(2)}(0)\|_F \cdot \|\Lambda_{(1),i}(t) \otimes \Lambda_{(1),i,\ell}(t) - \Lambda_{(1),i}(0) \otimes \Lambda_{(1),i,\ell}(0)\|_F\}$$

$$\leq O(\sqrt{L}R)$$

where the first step follows from simple algebra and Fact B.12, the second step follows from a basic tensor trick, the third step follows from simple algebra, triangle inequality, Part 4 of this lemma and $\|\Lambda_{(1),i,\ell}(t) - \Lambda_{(1),i,\ell}(0)\|_\infty \leq O(R/(Bd\sqrt{Ld}))$.

Hence, we have:

$$\|o_{(2),i}(t) - o_{(2),i}(0)\|_2 = \|\Lambda_{(1),i}(t)^\top \sigma_{(2),p}(t) - \Lambda_{(1),i}(0)^\top \sigma_{(2),p}(0)\|_2$$

$$\leq \max\{\|\Lambda_{(1),i}(t)\| \cdot O(RLdB), O(1) \cdot O(\sqrt{L}R)\}$$

$$\leq O(\sqrt{L}R),$$

where these steps follow from some basic algebra, Fact B.12 and Part 10 of this Lemma.

Here, we similarly apply Hoeffding inequality (Lemma B.14), and get: with a probability at least $1 - \delta$, we have:

$$\|\Lambda_{(2),i,\ell}(t) - \Lambda_{(2),i,\ell}(0)\|_2 \leq O(R).$$

By induction, we obtain the results of Part 11, 12, and 13.

**Proof of Part 14.** We have:

$$\mathcal{L}(t, \mathbb{D}) = \frac{1}{n}\|\mathsf{F}(t) - \mathsf{Y}\|_F^2$$

$$\leq L \max_{p \in [nL]} \|\mathsf{F}_p(t) - \mathsf{Y}_p\|_2^2$$

$$\leq L \max_{p \in [nL]} (\|\mathsf{F}_p(t)\|_2 + \|\mathsf{Y}_p\|_2)^2$$

$$\leq L \left(O(1) + O(\sqrt{d})\right)^2$$

$$\leq O(Ld),$$

where the first step follows from Lemma C.3, the second step follows from simple algebras, the third step follows from the Cauchy-Schwartz inequality, the last two steps follow from Definition B.3 and simple algebras.

**Proof of Part 15.** It is easy to prove:

$$\|\frac{\mathrm{d}\mathcal{L}(t, \mathbb{D})}{\mathrm{d}\mu_{(N)}(t)}\|_F \leq O(\sqrt{d}).$$

Besides, we can obtain:

$$\kappa U_{(N)}(t)\Lambda_{(N-1),i}(t)^\top (\operatorname{diag}(\mathbf{1}_n - \frac{1}{2}e_\ell)) \cdot \left(\operatorname{diag}(\sigma_{(N),p}(t)) - \sigma_{(N),p}(t)\sigma_{(N),p}(t)^\top\right) \Lambda_{(N-1),i}(t)$$

$$\leq O(\kappa L^2 dB)$$

$$\leq O(L^2 dB).$$

Then, by Hoeffding inequality (Lemma B.14), we have:

$$\|G_{(N-1),p}(t)\|_2^2 \leq \omega \cdot O(L^2 d^{2.5} B^3).$$

Choosing $\omega = o(\frac{1}{L^2 d^{2.5} B^3})$, we have: $\|G_{(N-1),p}(t)\|_2^2 \leq o(1)$.

By induction, we can show that: $\|\frac{\mathrm{d}\mathcal{L}(t,\mathbb{D})}{\mathrm{d}\mu_{(\nu)}(t)}\|_F \leq O(\sqrt{d})$.

Similarly, we have:

$$\|\frac{\mathrm{d}\mathcal{L}(t,\mathbb{D})}{\mathrm{d}\mu_{(\nu)}(t)}\|_F^2 \asymp \varepsilon^2 \cdot \mathcal{L}(t,\mathbb{D}).$$

**Proof of Part 16.** This proof is trivially similar with bounding $\|G_{(\nu),p}(t)\|_2^2 \leq o(1)$, then we choose $\kappa = 1/\sqrt{m}$ to meet the result.

This completes the proof of all parts of the lemma. $\qquad\square$

## E    KERNEL PERTURBATION

Here is a formal version of confirmation of Lemma 5.3:

**Lemma E.1** (Formal version of Lemma 5.3). *Assuming Assumption 5.1 and Definition 5.2 hold, denote the failure probability $\delta \in (0, 0.1)$, then the kernel perturbation bound is:*

$$\Pr\left[\lambda_{\min}(H_{(\nu)}(t)) < \lambda/2\right] < \delta.$$

*Therefore, bounding loss dynamics is given by ($C > 0$ is some constant):*

$$\Pr\left[\mathbb{E}[\frac{\mathrm{d}}{\mathrm{d}t}\mathcal{L}(t, \mathbb{D})] > -C \cdot \omega \lambda N \cdot \mathcal{L}(t, \mathbb{D})\right] < \delta.$$

*Proof.* **Proof of Part 1.**    First, we assume the condition $\|w_{(\nu),r}(t) - w_{(\nu),r}(0)\|_2 \leq R$ and $\|U_{(\nu)}(t) - U_{(\nu)}(0)\|_F \leq R$ as Lemma D.1, which will be confirmed as a provable property in the further analysis.

For any $p, q \in [nL]$, we have:

$$|H'_{(\nu),p,q}(t) - H'_{(\nu),p,q}(0)|$$

$$= \frac{1}{m}|o_{(\nu),p}(t)^\top o_{(\nu),p}(t) \sum_{r \in [m]} \mathbb{I}\{o_{(\nu),p}(t)^\top w_{(\nu),r}(t) > 0, o_{(\nu),q}(t)^\top w_{(\nu),r}(t) > 0\}$$

$$- o_{(\nu),p}(0)^\top o_{(\nu),p}(0) \sum_{r \in [m]} \mathbb{I}\{o_{(\nu),p}(0)^\top w_{(\nu),r}(0) > 0, o_{(\nu),q}(0)^\top w_{(\nu),r}(0) > 0\}|$$

$$\leq \frac{1}{m}(Q_{(\nu),p,q,1} + Q_{(\nu),p,q,2} + Q_{(\nu),p,q,3}),$$

where the first step follows from the definition of $H'_{(\nu),p,q}(t)$, and the second step follows from the triangle inequality and defining:

$$Q_{(\nu),p,q,1} := |o_{(\nu),p}(t)^\top o_{(\nu),p}(t) \sum_{r \in [m]} \mathbb{I}\{o_{(\nu),p}(t)^\top w_{(\nu),r}(t) > 0, o_{(\nu),q}(t)^\top w_{(\nu),r}(t) > 0\}$$

$$- o_{(\nu),p}(0)^\top o_{(\nu),p}(t) \sum_{r \in [m]} \mathbb{I}\{o_{(\nu),p}(t)^\top w_{(\nu),r}(t) > 0, o_{(\nu),q}(t)^\top w_{(\nu),r}(t) > 0\}|,$$

$$Q_{(\nu),p,q,2} := |o_{(\nu),p}(0)^\top o_{(\nu),p}(t) \sum_{r \in [m]} \mathbb{I}\{o_{(\nu),p}(t)^\top w_{(\nu),r}(t) > 0, o_{(\nu),q}(t)^\top w_{(\nu),r}(t) > 0\}$$

$$- o_{(\nu),p}(0)^\top o_{(\nu),p}(0) \sum_{r \in [m]} \mathbb{I}\{o_{(\nu),p}(t)^\top w_{(\nu),r}(t) > 0, o_{(\nu),q}(t)^\top w_{(\nu),r}(t) > 0\}|,$$

$$Q_{(\nu),p,q,3} := |o_{(\nu),p}(0)^\top o_{(\nu),p}(0) \sum_{r \in [m]} \mathbb{I}\{o_{(\nu),p}(t)^\top w_{(\nu),r}(t) > 0, o_{(\nu),q}(t)^\top w_{(\nu),r}(t) > 0\}$$

$$- o_{(\nu),p}(0)^\top o_{(\nu),p}(0) \sum_{r \in [m]} \mathbb{I}\{o_{(\nu),p}(0)^\top w_{(\nu),r}(0) > 0, o_{(\nu),q}(0)^\top w_{(\nu),r}(0) > 0\}|.$$

We assume a constant $R$ that satisfies $\|w_{(\nu),r}(t) - w_{(\nu),r}(0)\|_2 \leq R$ and $\|U_{(\nu)}(t) - U_{(\nu)}(0)\|_F \leq R$.

**Bounding $Q_{(\nu),p,q,1}$.** We have:

$$Q_{(\nu),p,q,1} \leq \left|\left(o_{(\nu),p}(t) - o_{(\nu),p}(0)\right)^\top o_{(\nu),q}(t)\right.$$

$$\left. \cdot \sum_{r=1}^m \mathbb{I}\{\langle o_{(\nu),p}(t), w_{(\nu),r}(t)\rangle > 0, \langle o_{(\nu),q}(t), w_{(\nu),r}(t)\rangle > 0\}\right|$$

$$\leq m\left|\left(o_{(\nu),p}(t) - o_{(\nu),p}(0)\right)^\top o_{(\nu),q}(t)\right|$$

$$\leq m\|o_{(\nu),p}(t) - o_{(\nu),p}(0)\|_2 \cdot \|o_{(\nu),q}(t)\|$$

$$\leq m \cdot O(\sqrt{L}R),$$

where the first step follows from the definition of $Q_{(\nu),p,q,1}$ and Cauchy-Schwarz inequality, the second step follows from $\mathbb{I}\{\langle o_{(\nu),p}(t), w_{(\nu),r}(t)\rangle > 0, \langle o_{(\nu),q}(t), w_{(\nu),r}(t)\rangle > 0\} \leq 1$, the third step follows from Cauchy-Schwarz inequality, the last step follows from Part 6 and Part 13 of Lemma D.1 and $\|o_{(\nu),q}(t)\| \leq O(1)$.

**Bounding $Q_{(\nu),p,q,2}$.** We omit this proof since it is similar to the proof of bounding $Q_{(\nu),p,q,1}$, we have:

$$Q_{(\nu),p,q,2} \leq m \cdot O(\sqrt{L}R).$$

**Bounding $Q_{(\nu),p,q,3}$.** We define the following event:

$$\mathsf{E}_{(\nu),p,r} := \{\exists w \in \mathbb{R}^d : \|w - w_{(\nu),r}(0)\|_2 \leq R,$$
$$\mathbb{I}\{\langle o_{(\nu),p}(0), w_{(\nu),r}(0)\rangle > 0\} \neq \mathbb{I}\{\langle o_{(\nu),p}(t), w_{(\nu),r}(t)\rangle > 0\}\}.$$

It is easy to hold that, once:

$$|\langle o_{(\nu),p}(0), w_{(\nu),r}(0)\rangle| \geq O(\sqrt{L}R) \cdot O(\sqrt{d}B) + O(1) \cdot R$$
$$\iff |\langle o_{(\nu),p}(0), w_{(\nu),r}(0)\rangle| \geq O(\sqrt{Ld}BR),$$

the event $\mathsf{E}_{(\nu),p,r}$ is false, since we combining Part 1, Part 9 and Part 13 of Lemma D.1, $\|o_{(\nu),q}(t)\| \leq O(1)$ and some simple algebra.

Following Fact B.10, we have:

$$\langle o_{(\nu),p}(0), w_{(\nu),r}(0)\rangle \sim \mathcal{N}(0, \|o_{(\nu),p}(0)\|_2^2).$$

We have:

$$\|o_{(\nu),p}(0)\|_2^2 = \sum_{\ell'=1}^{\ell} \sigma_{(\nu),p,\ell'}(t)^2 \|\Lambda_{(\nu),p,\ell'}(t)\|_2^2 = O(1) \cdot \sum_{\ell=1}^{L_i} \sigma_{(\nu),p,\ell'}(t)^2 \geq \ell \exp(-O(dB)) \geq \exp(-O(dB)),$$

where $\ell = p \bmod L$.

Then, the anti-concentration of $\langle o_{(\nu),p}(0), w_{(\nu),r}(0)\rangle$ shows that:

$$\Pr[\mathsf{E}_{(\nu),p,r}] \leq \frac{1}{\Theta(1) \cdot \|o_{(\nu),p}(0)\|_2} O(\sqrt{Ld}BR)$$
$$\leq O(\sqrt{L}R) \cdot \exp(O(dB)),$$

where the first step follows from Fact B.9, the second step follows from simple algebra and $\|o_{(\nu),p}(0)\|_2^2 \geq \exp(-O(dB))$.

We have:

$$\mathbb{E}[Q_{(\nu),p,q,3}] = \mathbb{E}\left[\left|o_{(\nu),p}(0)^\top o_{(\nu),q}(0)\right|\right.$$
$$\cdot \sum_{r=1}^{m}\left|\mathbb{I}\{\langle o_{(\nu),p}(t), w_{(\nu),r}(t)\rangle > 0, \langle o_{(\nu),q}(t), w_{(\nu),r}(t)\rangle > 0\}\right.$$
$$\left.\left. - \mathbb{I}\{\langle o_{(\nu),p}(0), w_{(\nu),r}(0)\rangle > 0, \langle o_{(\nu),q}(0), w_{(\nu),r}(0)\rangle > 0\}\right|\right]$$
$$\leq \mathbb{E}\left[O(1) \cdot \sum_{r=1}^{m}\left|\mathbb{I}\{\langle o_{(\nu),p}(t), w_{(\nu),r}(t)\rangle > 0, \langle o_{(\nu),q}(t), w_{(\nu),r}(t)\rangle > 0\}\right.\right.$$
$$\left.\left. - \mathbb{I}\{\langle o_{(\nu),p}(0), w_{(\nu),r}(0)\rangle > 0, \langle o_{(\nu),q}(0), w_{(\nu),r}(0)\rangle > 0\}\right|\right]$$
$$\leq O(1) \cdot \sum_{r=1}^{m} \mathbb{E}\left[\left|\mathbb{I}\{\langle o_{(\nu),p}(t), w_{(\nu),r}(t)\rangle > 0, \langle o_{(\nu),q}(t), w_{(\nu),r}(t)\rangle > 0\}\right.\right.$$

$$- \mathbb{I}\{\langle o_{(\nu),p}(0), w_{(\nu),r}(0)\rangle > 0, \langle o_{(\nu),q}(0), w_{(\nu),r}(0)\rangle > 0\}\Big|\Big]$$

$$\leq O(1) \cdot \sum_{r=1}^{m} \mathbb{E}\left[\mathbb{I}\{\mathsf{E}_{(\nu),p,r} \cup \mathsf{E}_{(\nu),q,r}\}\right]$$

$$\leq O(1) \cdot \sum_{r=1}^{m} \exp(O(dB)) \cdot R L^{\nu/2}$$

$$\leq m O(\sqrt{L}R) \cdot \exp(O(dB)),$$

where the first step follows from the definition of $Q_{(\nu),p,q,3}$, the second step follows from $\|o_{(\nu),p}(0)\|_2 \leq O(1)$, the third and fourth step follow from the rules of expectation, the last two steps follow from $\Pr[\mathsf{E}_{(\nu),p,r}] \leq O(\sqrt{L}R) \cdot \exp(O(dB))$ and simple algebra.

Hence, using Markov's inequality (Lemma B.15), with a probability at least $1 - \delta$, we have:

$$Q_{(\nu),p,q,3} \leq m \cdot O(\sqrt{L}R) \cdot \exp(O(dB))/\delta.$$

Combine the upper bounds on three terms, we have:

$$|H'_{(\nu),p,q}(t) - H'_{(\nu),p,q}(0)| \leq O(\sqrt{L}R) \cdot \exp(O(dB))/\delta.$$

We have:

$$\|H'_{(\nu)}(t) - H'_{(\nu)}(0)\|_F = \sqrt{\sum_{p=1}^{nL}\sum_{q=1}^{nL}(H_{(\nu),p,q}(t) - H_{(\nu),p,q}(0))^2}$$

$$\leq O(nL^{1.5}R) \cdot \exp(O(dB))/\delta.$$

Following Fact B.11 and choose

$$R \leq \frac{\omega\lambda\delta}{nL^{1.5}\exp(O(dB))}, \tag{5}$$

we have:

$$\lambda_{\min}(H'_{(\nu)}(t)) \geq \frac{3}{4}\omega\lambda.$$

It is trivial to have (Part 16 of Lemma D.1):

$$\|\gamma_{(\nu),p}(t)\|_2 \leq o(\frac{1}{\sqrt{m}}).$$

Therefore, we have:

$$|H_{(\nu),p,q}(t) - H'_{(\nu),p,q}(t)| = |\langle\gamma_{(\nu),p}(t), \gamma_{(\nu),q}(t)\rangle|$$

$$\leq o(1/m).$$

Furthermore, we have:

$$\|H_{(\nu)}(t) - H'_{(\nu)}(t)\|_F \leq o(\frac{n}{m}) \leq \omega\lambda/4,$$

where the last step follows from $m = \Omega(n/(\omega\lambda))$.

Similarly, following Fact B.11 and combining the previous result, we have:

$$\lambda_{\min}(H_{(\nu)}(t)) \geq \omega\lambda/2.$$

Following Fact B.13, we have:

$$\lambda_{\min}(H_{(\nu)}(t) \otimes I_d) \geq \omega\lambda/2.$$

Combining Lemma C.6, Part 15 of Lemma D.1 and Fact B.11, we can show that:

$$\mathbb{E}[\frac{\mathrm{d}}{\mathrm{d}t}\mathcal{L}(t,\mathbb{D})] \leq -C \cdot \omega\lambda N \cdot \varepsilon^2 \cdot \mathcal{L}(t,\mathbb{D}).$$

Solving this ODE, we have:

$$\mathbb{E}[\mathcal{L}(T,\mathbb{D}) \leq \exp(-C \cdot \varepsilon^2 \omega\lambda NT) \cdot \mathcal{L}(0,\mathbb{D}). \tag{6}$$

**Bounding Gradient Norm.** Following Part 3 of Lemma D.1, we have:

$$\max_{T\geq 0}\max_{\nu\in[N]}\max_{r\in[m]}\|w_{(\nu),r}(T) - w_{(\nu),r}(0)\|_2$$

$$\leq \max_{T\geq 0}\int_0^T \max_{\nu\in[N]}\max_{r\in[m]}\|\frac{\mathrm{d}\mathcal{L}(s,\mathbb{D})}{\mathrm{d}w_{(\nu),r}(s)}\|_2 \mathrm{d}s$$

$$\leq \max_{T\geq 0}\int_0^T \exp(-C \cdot \varepsilon^2\omega\lambda Ns) \cdot \mathcal{L}(0,\mathbb{D}) \cdot \frac{\omega}{\sqrt{m}} \cdot O(\sqrt{d}B)\mathrm{d}s$$

$$\leq \max_{T\geq 0}\int_0^T \exp(-C \cdot \varepsilon^2\omega\lambda Ns) \cdot \frac{1}{\sqrt{m}}\mathrm{d}s$$

$$= \frac{1}{\sqrt{m}}\max_{T\geq 0} -\frac{1}{C\varepsilon^2\omega\lambda N}\exp(-C \cdot \varepsilon^2\omega\lambda Ns)\Big|_{s=0}^{s=T}$$

$$\leq O(\frac{1}{\sqrt{m}\varepsilon^2\omega\lambda N}) \tag{7}$$

$$\leq R$$

where the first step follows from Eq. (2) and Cauchy-Schwartz inequality, the second step follows from Eq (6), the third step follows from Part 14 of Lemma D.1 and the choice of $\omega$, the last two steps follow from simple algebras and choosing

$$m = \Omega(\frac{n^2L^3\exp(O(dB))}{\varepsilon^4\omega^4\lambda^4\delta^2N^2}) \iff m/\mathrm{polylog}(N,m) = \Omega(\frac{n^2L^3\exp(Cd)}{\varepsilon^4\omega^4\lambda^4\delta^2N^2})$$

$$\iff m/\mathrm{polylog}(m) = \Omega(\frac{n^2L^3\exp(Cd)}{\varepsilon^4\omega^4\lambda^4\delta^2N^{\frac{4}{3}}})$$

$$\iff m^{\frac{2}{3}} = \Omega(\frac{n^2L^3\exp(Cd)}{\varepsilon^4\omega^4\lambda^4\delta^2N^{\frac{4}{3}}})$$

$$\iff m = \Omega(\frac{n^3L^5\exp(Cd)}{\varepsilon^6\omega^6\lambda^6\delta^3N^2})$$

where these steps follow from simple algebras.

Similarly, we have:

$$\max_{T\geq 0}\max_{\nu\in[N]}\|U_{(\nu)}(T) - U_{(\nu)}(0)\|_F \leq O(\frac{1}{\sqrt{m}\varepsilon^2\omega\lambda N}) \leq R$$

$\square$

# F    CONVERGENCE AND APPROXIMATION BOUND

## F.1    COMPLEXITY ANALYSIS

**Lemma F.1.** *We have*

- **Part 1.** *The time complexity (forward/backward) of the transformer on a single data point is* $O(NLmd) = O(\mathsf{M}L) \le O(\mathsf{M})$.

- **Part 2.** *The number of neurons in transformer is* $\mathsf{M} = O(N(md + d^2)) \le O(Nmd)$.

*Proof.* **Proof of Part 1.** We first note that the naive time complexity of matrix multiplication is $d_1 d_2 d_3$ for matrices $A \in \mathbb{R}^{d_1 \times d_2}$ and $B \in \mathbb{R}^{d_2 \times d_3}$. Formally, $\mathsf{Multiply}(A, B) = O(d_1 d_2 d_3)$.

The following analysis follows from $v \in [N]$, $p \in [nL]$ and $i = \lfloor p/L \rfloor$.

The complexity of $\mathsf{Multiply}(\Lambda_{(\nu-1),i}(t), U_{(\nu)}(t))$ is $O(Ld^2)$.

The complexity of $\mathsf{Multiply}(\Lambda_{(\nu-1),i}(t)U_{(\nu)}(t), \Lambda_{(\nu-1),i}(t)^\top)$ is $O(L^2 d)$.

The complexity of naive softmax is $O(L^2)$.

The complexity of $\mathsf{Multiply}(\Lambda_{(\nu-1),i}(t)^\top, \sigma_{(\nu),p}(t))$ is $O(Ld)$. Since the parallelization, total complexity should be $O(L^2 d)$.

The complexity of $\mathsf{Multiply}(W_{(\nu)}(t))^\top, o_{(\nu),p}(t))$ is $O(Lmd)$.

Since $A_{(\nu)} \in \{-1, +1\}^{m \times d}$, there exists a algorithm using addition operation to implement $\mathsf{Multiply}(A_{(\nu)}^\top, W_{(\nu)}(t))^\top o_{(\nu),p}(t)^\top)$ with complexity $O(Lmd)$.

Summing all complexity, we have the total complexity:

$$O(Ld^2 + L^2 d + L^2 + L^2 d + Lmd + Lmd) \le O(Lmd),$$

where this step follows from $m$ is the major term.

**Proof of Part 2.** This is trivial. $\qquad\square$

## F.2    TRAINING CONVERGENCE

**Theorem F.2** (Formal version of Theorem 5.5). *Assuming Assumption 5.1, Definition 5.4 and Definition 5.2 hold, denote the failure probability $\delta \in (0, 0.1)$, then with a probability at least $1 - \delta$, we have:*

- $(n, |\mathbb{B}|)$-**Fixed Convergence.** *Denote $\alpha = \mathrm{poly}(n, L, \exp(d), \frac{1}{\delta})$, we have:*

$$\mathcal{L}(\mathsf{T}, \mathbb{D}) \le \exp(-\frac{\varepsilon^2}{\alpha}\mathsf{M}\mathsf{T}).$$

- $(n, |\mathbb{B}|)$-**Dependent Convergence.** *Denote $\mathsf{N} = \Theta(\mathsf{C}^{\frac{1}{4}})$, $\mathsf{M} = \Theta(\mathsf{C}^{\frac{3}{4}})$, $\mathsf{T} = \Theta(\mathsf{N})$, we have:*

$$\mathcal{L}(\mathsf{N}, \mathbb{D}) \le \exp(-\varepsilon^2 \omega \mathsf{C}^{\frac{1}{4}}),$$

*Proof.* **Proof of $(n, |\mathbb{B}|)$-Fixed Convergence.** Following Lemma E.1, it is easy to have:

$$|\frac{\mathrm{d}}{\mathrm{d}t}\mathcal{L}(t, \mathbb{D})| \le O(\frac{N}{\sqrt{m}}) \cdot \mathcal{L}(t, \mathbb{D}) \le o(\frac{\sqrt{|\mathbb{B}|}}{\sqrt{n}B}\varepsilon^2 \omega \lambda N) \cdot \mathcal{L}(t, \mathbb{D}).$$

where the second step follows from $m = \Omega(n/(\varepsilon^2 \omega \lambda)^2)$ and $|\mathbb{B}| = \min_{t \ge 0} |\mathbb{B}(t)| \ge 1$ is the minimum batch size.

Then, following Hoeffding inequality (Lemma B.14), with a probability at least $1 - \delta$, we have:

$$|\frac{\mathrm{d}}{\mathrm{d}t}\mathcal{L}(t, \mathbb{D}) - \mathbb{E}[\frac{\mathrm{d}}{\mathrm{d}t}\mathcal{L}(t, \mathbb{D})]|$$

$$\leq o(\frac{\sqrt{|\mathbb{B}|}}{\sqrt{n}B}\varepsilon^2\omega\lambda N)\cdot\sqrt{\frac{n}{|\mathbb{B}|}\log(1/\delta)}\cdot\mathcal{L}(t,\mathbb{D})\leq o(\varepsilon^2\omega\lambda N)\cdot\mathcal{L}(t,\mathbb{D}).$$

Combining Lemma E.1, we have:

$$\frac{\mathrm{d}}{\mathrm{d}t}\mathcal{L}(t,\mathbb{D})\leq -C\varepsilon^2\omega\lambda N\cdot\mathcal{L}(t,\mathbb{D}).$$

Solving this ODE, we get:

$$\mathcal{L}(\mathsf{T},\mathbb{D})\leq\exp(-\frac{\varepsilon^2}{\alpha}\mathsf{M}\mathsf{T}),$$

where this step follows from $\omega=o(\frac{1}{\mathrm{poly}(L,d,\frac{1}{\delta})})$ and $m=\Omega(n,L,\exp(d),\frac{1}{\delta})$.

$(n,|\mathbb{B}|)$**-Dependent Convergence.** Similarly, we have:

$$\mathcal{L}(\mathsf{T},\mathbb{D})\leq\exp(-\varepsilon^2\omega N\cdot\frac{n}{|\mathbb{B}|})\leq\exp(-\varepsilon^2\omega\mathsf{N})=\exp(-\varepsilon^2\omega\mathsf{C}^{\frac{1}{4}}),$$

this inequality holds since $m$ is $\Omega(n^3)$ and $\omega=o(\frac{1}{NL^2d^{2.5}B^3})$ in Definition 5.2, $|\mathbb{B}|=\min_{t\geq 0}|\mathbb{B}(t)|\geq 1$, we choose $|\mathbb{B}|=1$ ($\mathsf{T}=\mathsf{N}$) and $\mathsf{C}=O(\mathsf{MN})=\Omega(\mathsf{N}^4)$. □

### F.3 APPROXIMATION

**Corollary F.3** (Formal version of Corollary 5.6)**.** *Assuming we have arbitrary dataset size* $\mathsf{N}\in(0,+\infty)$. *Let all scaling law factors be defined as Definition 5.4 and Assumption 5.1 and Definition 5.2 hold. Denote the failure probability* $\delta\in(0,0.1)$. *We define* $\alpha=O(Ld\sqrt{\log(1/\delta)})$. *For the Good Model Class* $\mathcal{F}_{\mathsf{M},\mathsf{T},\mathsf{N}}(\mathbb{D}')$ *with some* $\mathbb{D}'\subset\mathcal{X}\times\mathcal{Y}$ *and any function* $F':\mathcal{X}\to\mathcal{Y}$, *arbitrary error* $\epsilon>0$ *and compute cost* $\mathsf{C}$, *with a probability at least* $1-\delta$, *we have:*

$$\inf_{F\in\mathcal{F}_{\mathsf{M},\mathsf{T},\mathsf{N}}(\mathbb{D}'),\mathbb{D}'\subset\mathcal{X}\times\mathcal{Y}}\mathop{\mathbb{E}}_{(X,Y)\sim\mathcal{D}}[\|F(X)-F'(X)\|_F^2]\leq\epsilon,$$

*where* $\mathsf{C}=O(\mathsf{MN})=\Omega(256\alpha^8\epsilon^{-8}\wedge\varepsilon^{-8}\omega^4\log(2\epsilon^{-1})^4)$, $\mathsf{M}=\Omega(\mathsf{N}^3)$, $\mathsf{T}=\mathsf{N}$.

*Proof.* First, we have (Part 14 of Lemma D.1):

$$\mathcal{L}(t,\mathbb{D})\leq O(Ld).$$

We let $\mathbb{D}':=\{(X_i,F'(X_i)),X_i\sim\mathcal{X}\}_{i=1}^{\mathsf{N}}$, then $\mathcal{F}_{\mathsf{M},\mathsf{T},\mathsf{N}}(\mathbb{D}')=\{F(\cdot,\theta(\mathsf{T})),\theta(0)\sim\mathcal{N}(0,I_{\mathsf{M}}),\theta(\mathsf{T})\in\mathcal{A}_{\mathsf{T},\mathsf{N}}(\theta(0),\mathbb{D}')\}$, where we define $\mathcal{A}_{\mathsf{T},\mathsf{N}}(\theta(0),\mathbb{D}'):=\{\theta(0)+\int_0^T-\frac{\mathrm{d}}{\mathrm{d}\theta}\mathcal{L}(s,\mathbb{B}(s))\mathrm{d}s,\mathbb{B}(s)\subseteq\mathbb{D}'\}$. Following Theorem F.2, we have:

$$\mathcal{L}(\mathsf{N},\mathbb{D})\leq\exp(-\varepsilon^2\omega\mathsf{C}^{\frac{1}{4}}).$$

We denote $\mathcal{R}_{\mathcal{F}_{\mathsf{M},\mathsf{T},\mathsf{N}}(\mathbb{D})}(F):=\inf_{F\in\mathcal{F}_{\mathsf{M},\mathsf{T},\mathsf{N}}(\mathbb{D})}\mathbb{E}_{(X,Y)\sim\mathcal{D}}[\|F(X)-F'(X)\|_F^2]$

Then we apply Hoeffding inequality (Lemma B.14) to each data point, we have:

$$|\mathcal{R}_{\mathcal{F}_{\mathsf{M},\mathsf{T},\mathsf{N}}(\mathbb{D})}(F)-\mathcal{L}(\mathsf{N},\mathbb{D})|\leq O\left(\frac{Ld}{\sqrt{\mathsf{N}}}\sqrt{\log(1/\delta)}\right). \tag{8}$$

Therefore, we have:

$$\mathcal{R}_{\mathcal{F}_{\mathsf{M},\mathsf{T},\mathsf{N}}(\mathbb{D})}(F)\leq O\left(\frac{Ld}{\sqrt{\mathsf{N}}}\sqrt{\log(1/\delta)}\right)+\mathcal{L}(\mathsf{N},\mathbb{D})$$

$$\leq\frac{\alpha}{\sqrt{\mathsf{N}}}+\exp(-\varepsilon^2\omega\mathsf{C}^{\frac{1}{4}}),$$

where $\alpha=O(Ld\sqrt{\log(1/\delta)})$.

Then we choose $\mathsf{C}=\Omega(256\alpha^8\epsilon^{-8}\wedge\varepsilon^{-8}\omega^4\log(2\epsilon^{-1})^4)$ and following Theorem F.2, then $\mathsf{M}=\Omega(\mathsf{N}^3)$, $\mathsf{T}=\mathsf{N}$.

□

## G  SCALING LAW

### G.1  GENERALIZATION BOUND OF EMPIRICAL RISK MINIMIZER

**Definition G.1** (Van Der Vaart & Wellner (1996) and Yang & Barron (1999))**.** *For a metric space $(\mathcal{S}, \mathrm{d})$ and $\varepsilon > 0$, a finite set $\mathcal{S}' \subset \overline{\mathcal{S}}$ is called $\varepsilon$-covering if for any $x \in \mathcal{S}$ there exists $y \in \mathcal{S}'$ such that $\mathrm{d}(x, y) \leq \varepsilon$, and the logarithm of the minimum cardinality of $\varepsilon$-covering is called covering $\varepsilon$-entropy and denoted by $V_{(\mathcal{S}, \mathrm{d})}(\varepsilon)$. Here, $\overline{\mathcal{S}}$ is the completion of $\mathcal{S}$ with respect to the metric $\mathrm{d}$.*

**Lemma G.2** (Lemma 11 of Schmidt-Hieber (2017), modified by Theorem 2.1 in Hayakawa (2019) and in the setting of this paper)**.** *We denote $\mathcal{F}_{\mathsf{M},\mathsf{T},\mathsf{N}} = \cup_{\mathbb{D} \subset \mathcal{D}} \mathcal{F}_{\mathsf{M},\mathsf{T},\mathsf{N}}(\mathbb{D})$. Then for any $\mathbb{D} \subset \mathcal{D}$, let $F$ be the empirical risk minimizer taking values in $\mathcal{F}_{\mathsf{M},\mathsf{T},\mathsf{N}}$. Suppose every element $F \in \mathcal{F}_{\mathsf{M},\mathsf{T},\mathsf{N}}$ satisfies $\mathrm{d}(F) := \max_{\ell \in [L]} \|F_\ell\|_{L^\infty(\mathcal{X})} \leq B_F$ for some fixed $B_F > 0$. Then, for an arbitrary $\varepsilon > 0$, if $V_{(\mathcal{F}, \mathrm{d})}(\varepsilon) \geq 1$, then*

$$\|F - F^*\|_{L^F(\mathcal{X})}^2 \leq 4 \inf_{F' \in \mathcal{F}_{\mathsf{M},\mathsf{T},\mathsf{N}}} \|F' - F^*\|_{L^F(\mathcal{X})}^2 + L \cdot O\left(\frac{B_F^2 V_{(\mathcal{F}_{\mathsf{M},\mathsf{T},\mathsf{N}}, \mathrm{d})}(\varepsilon)}{n} + B_F \varepsilon\right).$$

**Lemma G.3.** $\delta \in (0, 0.1)$. *With a probability at least $1 - \delta$, we have:*

- **Part 1.** $\max_{\ell \in L} \|F_\ell\|_{L^\infty(\mathcal{X})} \leq O(\sqrt{\varepsilon}) =: B_F$.

- **Part 2.** *For any constant $\varepsilon > 0$, we can show that $1 \leq V_{(\mathcal{F}, \mathrm{d})}(\varepsilon) \leq O(d \cdot \log(1/\sqrt{\varepsilon}))$.*

- **Part 3.** *For any $F' \in \mathcal{F}_{\mathsf{M}',\mathsf{T}',\mathsf{N}'}$, we have:*

$$\sup_{X \in \mathcal{X}} \inf_{F \in \mathcal{F}} \|F(X) - F'(X)\|_F^2 \leq O\left(\frac{1}{\mathsf{C}^{\frac{3}{8}} \omega}\right)$$

*Proof.* **Proof of Part 1.** Following Part 5 of Lemma D.1 and Eq. (1), we have:

$$\max_{\ell \in [L], X \in \mathcal{X}} \|F_\ell(X)\|_2^2 \leq O(\varepsilon).$$

We have:

$$\|F_\ell\|_{L^\infty(\mathcal{X})} \leq \left(\max_{\ell \in [L]} \int_{\mathcal{X}} \|F_\ell(X)\|_\infty \mathrm{d}X\right)^{\frac{1}{2}}$$

$$\leq \left(\frac{1}{L} \mathrm{Volume}(\mathcal{X}) \cdot O(\varepsilon)\right)^{\frac{1}{2}}$$

$$\leq \left(\frac{\Theta(1)^{\frac{d}{2}}}{\frac{d}{2}!} \cdot O(\varepsilon)\right)^{\frac{1}{2}} \leq O(\sqrt{\varepsilon}),$$

where the first step follows from the $L^\infty$ norm, the second step follows from $\max_{\ell \in [L], X \in \mathcal{X}} \|F_\ell(X)\|_2^2 \leq O(\varepsilon)$, the third step follows from $\mathcal{X}$ is a $d$-dimensional ball with radius $\Theta(1)$, the last step follows from simple algebras.

**Proof of Part 2.** Following Part 1 of this Lemma, we have:

$$\sup_{F, F' \in \mathcal{F}_{\mathsf{M},\mathsf{T},\mathsf{N}}} \max_{\ell \in L} \|F_\ell - F'_\ell\|_{L^\infty(\mathcal{X})} \leq \sup_{F, F' \in \mathcal{F}_{\mathsf{M},\mathsf{T},\mathsf{N}}} \max_{\ell \in L} (\|F_\ell\|_{L^\infty(\mathcal{X})} + \|F'_\ell\|_{L^\infty(\mathcal{X})}) \leq 2B_F,$$

where the first step follows from simple algebras. Then we have the logarithm of the covering number:

$$V_{(\mathcal{F}_{\mathsf{M},\mathsf{T},\mathsf{N}}, \mathrm{d})}(\varepsilon) = \log\left(\left(\frac{2B_F}{\varepsilon}\right)^d\right) \leq O(d \cdot \log(1/\sqrt{\varepsilon})).$$

**Proof of Part 3.** We choose $\mathsf{C} = \mathsf{C}' = \Omega(256\alpha^8 \epsilon^{-8} \wedge \varepsilon^{-8} \omega^4 \log(2\epsilon^{-1})^4)$, $\mathsf{M} = \mathsf{M}'$, $\mathsf{T} = \mathsf{T}$ and $\mathsf{N} = \mathsf{N}'$ as Corollary F.3, let $N' = \Theta(1)$, $\mathsf{M}' = N'^3$, then we have:

$$m' = \Theta(\mathsf{C}'^{\frac{3}{4}}) = \Omega(64\alpha^6 \epsilon^{-6} \wedge \varepsilon^{-6} \omega^3 \log(2\epsilon^{-1})^3).$$

Next,

$$\sup_{X \in \mathcal{X}} \inf_{F \in \mathcal{F}_{\mathsf{M,T,N}}} \|F(X) - F'(X)\|_F^2 \leq O(\varepsilon^2 R),$$

where this step follows from Part 11 of Lemma D.1 and Eq (1).

**Relax** $R$. We combine Eq. (5) and Eq (7) to relax the bound on $R$, we get:

$$R = \frac{1}{\sqrt{m}\varepsilon^2 \omega \lambda N} \leq \frac{\omega \lambda \delta}{n L^{1.5} \exp(O(dB))}.$$

Hence, we show:

$$\sup_{X \in \mathcal{X}} \inf_{F \in \mathcal{F}_{\mathsf{M,T,N}}} \|F(X) - F'(X)\|_F^2 \leq O(\frac{1}{\mathsf{C}^{\frac{3}{8}} \omega}),$$

where this step follows that $\lambda$ is some fixed constant (Assumption 5.1) and $N = \Theta(1)$ and $\mathsf{M} = \mathsf{C}^{\frac{3}{4}}$ (Corollary F.3). $\qquad\square$

## G.2 GENERAL PRETRAINING SCALING LAW

**Theorem G.4** (Formal version of Theorem 6.1)**.** *Let all pre-conditions hold as Corollary F.3. For any $\varepsilon \in (0,1)$, with a probability at least $1 - \delta$, there exists:*

$$\inf_{\mathcal{F}_{\mathsf{M,T,N}}(\mathbb{D})} \sup_{\mathbb{D} \in \mathcal{D}} \Delta\mathcal{R}(F) \leq \begin{cases} O(\frac{\alpha^2}{\omega \mathsf{C}^{\frac{3}{8}}}) + O(\frac{\varepsilon \cdot d \cdot \log(1/\sqrt{\varepsilon})}{\mathsf{C}^{\frac{1}{4}}}) + \varepsilon^{\frac{3}{2}}, & \varepsilon \geq \sqrt{\frac{\log(\mathsf{C}^{\frac{1}{8}}/\alpha)}{\mathsf{C}^{\frac{1}{4}}\omega}} \\ \exp(-\varepsilon^2 \omega \mathsf{C}^{\frac{1}{4}}) + O(\frac{\varepsilon \cdot d \cdot \log(1/\sqrt{\varepsilon})}{\mathsf{C}^{\frac{1}{4}}}) + \varepsilon^{\frac{3}{2}}, & otherwise \end{cases},$$

*note that* $\mathsf{M} = \Omega(\mathsf{N}^3)$, $\qquad \mathsf{T} = \mathsf{N}$, $\qquad \mathsf{C} = O(\mathsf{MN}) = \Omega(\mathsf{N}^4)$.

*Proof.* We have:

$$\begin{aligned}
\Delta\mathcal{R}(F) &= \mathcal{R}(F) - \mathcal{R}(F^*) \\
&= \mathbb{E}_{(X,Y) \sim \mathcal{D}}[\|F(X) - Y\|_F^2 - \|F^*(X) - Y\|_F^2] \\
&= \mathbb{E}_{(X,Y) \sim \mathcal{D}}[\|F(X) - F^*(X) - \Xi\|_F^2 - \|\Xi\|_F^2] \\
&= \mathbb{E}_{(X,Y) \sim \mathcal{D}}[\|F(X) - F^*(X)\|_F^2 - \mathrm{vec}(F(X) - F^*(X))^\top \mathrm{vec}(\Xi) + \|\Xi\|_F^2 - \|\Xi\|_F^2] \\
&= \mathbb{E}_{(X,Y) \sim \mathcal{D}}[\|F(X) - F^*(X)\|_F^2] \\
&= \frac{1}{\mathrm{Volume}(\mathcal{X})} \|F - F^*\|_{L^F(\mathcal{X})}^2 \\
&\leq \Theta(\frac{1}{L}) \cdot \|F - F^*\|_{L^F(\mathcal{X})}^2
\end{aligned}$$

where the first step follows from the definitions of $\mathcal{R}$ and $\Delta\mathcal{R}$, the second step follows from $Y = F^*(X) + \Xi$, the third step follows from simple algebras, the fourth step follows from $\mathbb{E}[\Xi] = \mathbf{0}_{L \times d}$, the fifth step follows from the definition of $L^F$ norm, the last step follows from simple algebras that $\mathcal{X}$ is a space summing $L$ balls.

**Bounding Approximation Error.** We have:

$$\inf_{F \in \mathcal{F}_{\mathsf{M,T,N}}} \|F - F^*\|_{L^F(\mathcal{X})}$$

$$\leq \inf_{F' \in \mathcal{F}_{\mathsf{M,T,N}}(\mathbb{D}'), \mathbb{D}' \subset \mathcal{X} \times \mathcal{Y}} \|F'(X) - F^*(X)\|_{L^F(\mathcal{X})} + \inf_{F \in \mathcal{F}_{\mathsf{M,T,N}}} \|F(X) - F'(X)\|_{L^F(\mathcal{X})}$$

$$\leq \mathbb{E}_{(X,Y) \sim \mathcal{D}}[\inf_{F' \in \mathcal{F}_{\mathsf{M,T,N}}(\mathbb{D}'), \mathbb{D}' \subset \mathcal{X} \times \mathcal{Y}} \|F'(X) - F^*(X)\|_F + \inf_{F \in \mathcal{F}_{\mathsf{M,T,N}}} \|F(X) - F'(X)\|_F]$$

$$\leq \mathbb{E}_{(X,Y) \sim \mathcal{D}}[\inf_{F' \in \mathcal{F}_{\mathsf{M,T,N}}(\mathbb{D}'), \mathbb{D}' \subset \mathcal{X} \times \mathcal{Y}} \|F'(X) - F^*(X)\|_F] + O(\frac{1}{\mathsf{C}^{\frac{3}{16}} \sqrt{\omega}})$$

$$\leq \sqrt{\epsilon} + O(\frac{1}{\mathsf{C}^{\frac{3}{16}}\sqrt{\omega}}),$$

where the first step follows from Cauchy-Schwartz inequality, the second step follows that minima is smaller than the average, the third step follows from the Part 3 of Lemma G.3, the last step follows from Corollary F.3.

*Condition 1.* When $\varepsilon \geq \sqrt{\frac{\log(\mathsf{C}^{\frac{1}{8}}/\alpha)}{\mathsf{C}^{\frac{1}{4}}\omega}}$, we have:

$$\epsilon = O(\frac{\alpha}{\mathsf{C}^{\frac{1}{8}}}).$$

Then,

$$\inf_{F \in \mathcal{F}_{\mathsf{M,T,N}}} \|F - F^*\|_{L^F(\mathcal{X})} \leq O(\frac{\alpha}{\mathsf{C}^{\frac{1}{16}}}) + O(\frac{1}{\mathsf{C}^{\frac{3}{16}}\sqrt{\omega}}) \leq O(\frac{\alpha}{\mathsf{C}^{\frac{1}{16}}\sqrt{\omega}}).$$

*Condition 2.* When $\varepsilon \leq \sqrt{\frac{\log(\mathsf{C}^{\frac{1}{8}}/\alpha)}{\mathsf{C}^{\frac{1}{4}}\omega}}$, we have:

$$\epsilon = \exp(-\varepsilon^2 \omega \mathsf{C}^{\frac{1}{4}}),$$

where this step follows from $\varepsilon = o(\mathsf{N}^{-2})$ and $\mathsf{N} = \mathsf{C}^{\frac{1}{4}}$.

Then,

$$\inf_{F \in \mathcal{F}_{\mathsf{M,T,N}}} \|F - F^*\|_{L^F(\mathcal{X})} \leq \exp(-\varepsilon^2 \omega \mathsf{C}^{\frac{1}{4}}) + O(\frac{1}{\mathsf{C}^{\frac{3}{16}}\sqrt{\omega}}) \leq \exp(-\varepsilon^2 \omega \mathsf{C}^{\frac{1}{4}}),$$

where the second inequality follows from $\exp(-\varepsilon^2 \omega \mathsf{C}^{\frac{1}{4}}) \leq O(\frac{\alpha}{\mathsf{C}^{\frac{1}{8}}})$ when $\varepsilon \leq \sqrt{\frac{\log(\mathsf{C}^{\frac{1}{8}}/\alpha)}{\mathsf{C}^{\frac{1}{4}}\omega}}$.

**Bounding Generalization Error.** We have:

$$B_F \varepsilon \leq \varepsilon^{\frac{3}{2}},$$

where this step follows from Part 1 of Lemma G.3.

Next,

$$\frac{B_F^2 V_{(\mathcal{F}_{\mathsf{M,T,N}},\mathrm{d})}(\varepsilon)}{\mathsf{N}} \leq O(\frac{\varepsilon \cdot d \cdot \log(1/\sqrt{\varepsilon})}{\mathsf{N}}) \leq O(\frac{\varepsilon \cdot d \cdot \log(1/\sqrt{\varepsilon})}{\mathsf{C}^{\frac{1}{4}}}),$$

where the first step follows from Part 1 and 2 of Lemma G.3.

**Result.** We combine all results above and Theorem G.2, obtaining:

$$\inf_{\mathcal{F}_{\mathsf{M,T,N}}(\mathbb{D})} \sup_{\mathbb{D} \in \mathcal{D}} \Delta \mathcal{R}(F) \leq \begin{cases} O(\frac{\alpha^2}{\omega \mathsf{C}^{\frac{1}{8}}}) + O(\frac{\varepsilon \cdot d \cdot \log(1/\sqrt{\varepsilon})}{\mathsf{C}^{\frac{1}{4}}}) + \varepsilon^{\frac{3}{2}}, & \varepsilon \geq \sqrt{\frac{\log(\mathsf{C}^{\frac{1}{8}}/\alpha)}{\mathsf{C}^{\frac{1}{4}}\omega}} \\ \exp(-\varepsilon^2 \omega \mathsf{C}^{\frac{1}{4}}) + O(\frac{\varepsilon \cdot d \cdot \log(1/\sqrt{\varepsilon})}{\mathsf{C}^{\frac{1}{4}}}) + \varepsilon^{\frac{3}{2}}, & \text{otherwise} \end{cases}.$$

$\square$

## G.3 Upper Bound

**Theorem G.5** (Formal version of Theorem 6.2). *Let all pre-conditions hold as Theorem 6.1. For any choice of $\varepsilon$, we let $\mathsf{C} = \Omega(\varepsilon^{-12})$ to offset the negative effect of the grokking coefficient $\varepsilon$, thus,*

$$\Pr\left[\inf_{\mathcal{F}_{\mathsf{M,T,N}}(\mathbb{D})} \sup_{\mathbb{D} \in \mathcal{D}} \Delta \mathcal{R}(F) \asymp O(\frac{\alpha}{\mathsf{C}^{\frac{1}{8}}})\right] \geq 1 - \delta.$$

*Proof.* We follow Eq. (8), for all $\mathbb{D} \in \mathcal{D}$, we can show that:

$$\inf_{F \in \mathcal{F}_{\mathsf{M},\mathsf{T},\mathsf{N}}(\mathbb{D})} \mathcal{R}(F) \leq \frac{\alpha}{\sqrt{\mathsf{N}}} - \exp(-\varepsilon^2 \omega \mathsf{C}^{\frac{1}{4}})$$

$$\leq \frac{\alpha}{\sqrt{\mathsf{N}}} - \exp(-\omega \mathsf{C}^{\frac{1}{12}})$$

$$\leq O(\frac{\alpha}{\sqrt{\mathsf{N}}}) \leq O(\frac{\alpha}{\mathsf{C}^{\frac{1}{8}}}),$$

where these steps follow from simple algebras and $\mathsf{N} = o(\mathsf{C}^{-4})$.

At the same time, the upper bound is trivial.

$\square$

## H  LLMs Usage Disclosure

LLMs were used only to polish language, such as grammar and wording. These models did not contribute to idea creation or writing, and the authors take full responsibility for this paper's content.

