# OpenReview forum: "On the Scaling Theory of Multi-Layer Transformers"
_ICLR.cc/2026/Conference — Submitted to ICLR 2026_

### Official Review · Reviewer_CH4M · 2025-10-30

**Soundness:** 3
**Presentation:** 3
**Contribution:** 3
**Rating:** 4
**Confidence:** 3

**Summary:**

This paper provides a theoretical framework for analyzing the scaling laws in LLMs by modeling the dynamics of a multi-layered transformer as an ODE. Then, using this framework and assuming one pass SGD training on sequence-to-sequence data and kernel behavior of LLMs, bounds are provided for excess loss risk during training. The obtained bound gives an excess risk bound in terms of the compute C, as well as shows a phase transition in the obtained upper bound that depends on the grokking coefficient epsilon.

**Strengths:**

- Unlike prior work working on toy settings, this work successfully models multi-layered transformers training dynamics using SGD by formulating it using kernels and using NTK. Even though there are assumptions to frame multi-layered transformers as an ODE, the obtained insights that is applicable for large scale transformers is useful
- The upper bound shows phase transition and also links it to the grokking coefficient at initialization, and is also validated empirically.

**Weaknesses:**

- In Def 5.2, the constraint on w seems too small, also m is too large, scaling exponentially with d. Can the authors please provide evidence from practical multi-layered transformers to check if these are practical assumptions?
- The theory is based on decoder-only transformers (line 71) but all experiments are on vision transformers, which are not decoder only transformers. Can you please clarify if the theory holds for vision transformers and how or provide experiments with decoder only LLMs?
- Scaling rate C^{-1/8} is very slow compared to empirical scaling rate such as in Chinchilla paper. To that end, the phase transition observed in the bound, could that be just a property of the bound and not necessarily LLM behavior (which has different scaling rate than the bound?)
- Grokking coefficient: Could the authors point to initialization schemes that uses the Grokking coefficient, I am not aware of this being a standard practice like Xavier/Kaiming initializations.

**Questions:**

Please see the weaknesses section for questions

---

### Official Review · Reviewer_6JtY · 2025-11-01

**Soundness:** 3
**Presentation:** 3
**Contribution:** 2
**Rating:** 6
**Confidence:** 2

**Summary:**

This paper provides the first rigorous theoretical analysis of scaling laws for multi-layer transformer-based language models, establishing convergence guarantees for generalization error with a rate of $\Theta(C^{-1/8})$ where $C$ is computational cost. The authors formalize training dynamics as an ODE system under the Neural Tangent Kernel regime and derive a three-stage scaling theory that transitions from classical scaling to exponential convergence (grokking phenomenon) and finally to data-limited regime. Experimental validation on image classification tasks demonstrates alignment between theoretical predictions and empirical scaling behavior.

I am not an expert for such a theoretical paper.

**Strengths:**

1. The paper builds on multi-layer transformers that is not toy setting as previous works.
2. This paper has a good and rigorious theory induction. Moreover, this theory explains grokking phenomenon to some degree. Some empirical experiments are also conducted for validation.

**Weaknesses:**

1. The proven rate of Θ(C^{-1/8}) is relatively slow compared to empirical scaling laws (e.g., Kaplan et al. 2020 report power laws closer to $C^{-0.5}$). Moreover, I think the coefficient here (-1/8)  is variable with dataset, experimental settings in real-world LLM pretraining. How would the coefficient be fixed?

2. Some settings could be still far away from real-world LLM training, and I think some of them including
- a. AdamW vs. SGD optimizer
- b. Positional Encoding: RoPE vs. NoPE
- c. Pretraining on very large and diverse datasets vs. Finetuning on small datasets like MNIST, CIFAR-10/100

I know it's hard and this paper has advanced one step. But these inconsisitency could still lead to unexpected gaps between theory and real-world practice.

**Questions:**

See above please.

---

### Official Review · Reviewer_fQgs · 2025-11-02

**Soundness:** 3
**Presentation:** 2
**Contribution:** 2
**Rating:** 2
**Confidence:** 4

**Summary:**

The paper studies the scaling laws of attention based models extending the analysis in the kernel regime calculations from the deep networks. It derives a

**Strengths:**

The paper derives the convergence and scaling laws in the kernel regime for transformers, it extends the convergence and generalization from the deep networks using NTK and derive kernel based bounds and rates.

The works goes beyond toy models for laying foundations of large language models.

With certains assumptions on the initial scales of the model, width of the MLPs, the paper dervies a scaling law for the convergence of excess risk.

**Weaknesses:**

The lazy regime does not correspond to the practical learning setup which operated in rich or feature learning regime. Hence, the results derived are not broadly applicable.

It would be nice to get more empirical validation for the scaling law the paper derives beyond vision transformers.

**Questions:**

Q1) I do not completely understand the distinction between the over-parameterized and data limited regimes, what is the precise difference

Q2) How is the convergence speed exponential in the over-parameterized regime, isn't it still bounded by $O(1/C^{1/8})$, also it is hard to decipher how this phase corresponds to grokking and faster generalization speed.

Q3) In the proof sketch of theorem 6.1, the authors cite Lemma 11 of Schmidt-Hieber 2017 which do not exist, this citation needs to be revisited.

---

### Meta-Review · Area_Chair_P1vA · 2025-12-29

**Summary:**

## Summary
This submission analyzes the training dynamics of decoder-only transformer-based language models using the one-pass SGD. The authors adopt a kernel-based analytical framework (i.e., lazy training) to investigate the scaling behavior. Under the kernel/lazy training regime, the authors analyze how excess risk scales as a function of compute $C$.

## Reviewer Concerns
- **relevance of lazy training**. Reviewers pointed out that the conditions in Def 5.2 are too restrictive, and the lazy regime does not correspond to practical learning setups.
- **gap to practical setups**. There are other gaps between theory and practice, such as the slower-than-observed scaling exponent 1/8, the mismatch of decoder-only analysis and ViT experiments, and others (use of grokking coefficient, AdamW vs SGD)
- **readability**. Reviewer fQgs raised some readability issues as well, e.g., unclear distinction between over-parameterized and data-limited regimes and unclear connection of phase transitions to grokking.

## Overall Assessment
I do concur with reviewers’ concerns. Unfortunately, the authors did not respond to the reviews and the issues remain unresolved; therefore, I cannot recommend acceptance of this paper at this time.

**Reviewer Concerns:**

The authors did not submit their responses, hence all concerns remain unresolved.

**Reviewer Scores:**

The authors did not submit their responses, hence there would have been no change in the review scores.

---

### Decision · Program_Chairs · 2026-01-26

Reject